# Untrained Neural Nets for Snapshot Compressive Imaging: Theory and Algorithms

**Mengyu Zhao**[1], **Xi Chen**[1], **Xin Yuan**[2], **Shirin Jalali**[1]*
[1] ECE Department, Rutgers University, New Brunswick
[2] School of Engineering, Westlake University

## Abstract

Snapshot compressive imaging (SCI) recovers high-dimensional (3D) data cubes from a single 2D measurement, enabling diverse applications like video and hyperspectral imaging to go beyond standard techniques in terms of acquisition speed and efficiency. In this paper, we focus on SCI recovery algorithms that employ untrained neural networks (UNNs), such as deep image prior (DIP), to model source structure. Such UNN-based methods are appealing as they have the potential of avoiding the computationally intensive retraining required for different source models and different measurement scenarios. We first develop a theoretical framework for characterizing the performance of such UNN-based methods. The theoretical framework, on the one hand, enables us to optimize the parameters of data-modulating masks, and on the other hand, provides a fundamental connection between the number of data frames that can be recovered from a single measurement to the parameters of the untrained NN. We also employ the recently proposed bagged-deep-image-prior (bagged-DIP) idea to develop SCI Bagged Deep Video Prior (SCI-BDVP) algorithms that address the common challenges faced by standard UNN solutions. Our experimental results show that in video SCI our proposed solution achieves state-of-the-art among UNN methods, and in the case of noisy measurements, it even outperforms supervised solutions. Code is publicly available at `https://github.com/Computational-Imaging-RU/SCI-BDVP`.

## 1 Introduction

Snapshot Compressive Imaging (SCI) refers to imaging systems that optically encode a three-dimensional (3D) data cube into a two-dimensional (2D) image and computationally recover the 3D data cube from the 2D projection. As a novel approach in computational imaging, SCI has attracted significant attention in recent years. Initially proposed for spectral imaging [1], its application has since expanded to various fields, including video recording [2], depth imaging [3], and coherence tomography [4] (Refer to [5] for a comprehensive review).

The key advantage of SCI systems lies in significantly accelerating the data acquisition process. Traditional hyperspectral imaging methods, for example, often encounter bottlenecks due to their reliance on spatial or wavelength scanning, leading to time-consuming operations. In contrast, hyperspectral SCI systems capture measurements across multiple pixels and wavelengths in a single snapshot, effectively bypassing this limitation [6].

The optical encoding process in SCI systems can be mathematically modeled as a linear measurement system, characterized by a sparse and structured sensing matrix, commonly referred to as a 'mask'. Consequently, SCI recovery algorithms aim to reconstruct high-dimensional (HD) 3D data from a

---

*Shirin Jalali is the corresponding authors <shirin.jalali@rutgers.edu>.

highly underdetermined system of linear equations. A wide range of SCI recovery methods has been proposed in the literature, which can broadly be categorized into:

**Classic approaches:** These methods model source structure using convex regularization functions and employ convex optimization techniques (e.g., [7, 8, 9, 10]). While robust to measurement and source distribution non-idealities, they are typically limited to simpler structures and challenging to extend to 3D HD data cubes central to SCI applications. **DNN-based methods:** These approaches use deep neural networks (DNNs) to capture complex source structures, learning from training data. They can be further categorized as: i) End-to-end solutions (e.g., [11, 12, 13, 14, 15, 16]); ii) Iterative plug-and-play solutions (e.g., [17, 18]); iii) Unrolled methods (e.g., [19, 20, 21, 22, 23, 24]). While these methods extend beyond simple structures to model intricate source patterns, they require extensive training data, often struggle with generalization, and are computationally intensive.

An alternative approach to SCI recovery involves using UNNs, such as deep image prior (DIP) [25] or deep decoder [6], to model the source structure. These methods capture complex source structures without requiring any training data. Existing UNN-based SCI solutions either recover the image end-to-end in one shot [26] or employ iterative methods akin to projected gradient descent (PGD) [27]. Despite their advantages, these approaches often exhibit lower performance compared to pre-trained methods and may require additional data processing steps for enhancement.

In this work, we focus on leveraging UNNs to address the SCI problem. We begin by establishing a theoretical framework for analyzing UNN-based methods, providing insights into optimizing the adjustable SCI masks, under both noise-free and noisy measurements. We then explore DIP-based algorithms and introduce SCI-BDVP solutions. Our results demonstrate the robustness of these solutions to measurement noise and their competitive performance across diverse datasets, using a consistent set of parameters.

## 1.1 Contributions of this Work

**Theoretical:** We theoretically characterize the performance of DIP-based SCI recovery optimization for both noise-free and noisy measurements. Using our theoretical results, we establish an upper bound on the number of frames that can be recovered from a single 2D measurement, as a function of the dimensions of the DIP. Furthermore, we show how the developed theoretical results enable us to optimize the parameters of the masks for both noisy and noise-free cases, enhancing the performance of the recovery process.

**Algorithmic:** Inspired by the newly proposed bagged-DIP algorithm for the problem of coherent imaging [28], developed to address common shortcomings of DIP-based solutions for inverse problems, we explore the application of bagged-DIP for SCI recovery. We conduct extensive experimental evaluations, demonstrating the following: **i)** Confirmation of our theoretical results on the optimized masks for both noise-free and noisy measurements. **ii)** The proposed SCI-BDVP solution robustly achieves state-of-the-art performance among UNN-based solutions in the case of noise-free measurements. **iii)** In scenarios with noisy measurements, our proposed method achieves state-of-the-art performance among both end-to-end supervised and untrained methods.

## 1.2 Notations

Vectors are represented by bold characters like $\mathbf{x}$ and $\mathbf{y}$. $\|\mathbf{x}\|_2$ denotes the $\ell_2$ norm of $\mathbf{x}$. For $\mathbf{X} \in \mathbb{R}^{n_1 \times n_2}$, $\text{Vec}(\mathbf{X}) \in \mathbb{R}^n$ denotes the vectorized version of $\mathbf{X}$, where $n = n_1 n_2$. This vector is created by concatenating the columns of $\mathbf{X}$. Given $\mathbf{A}, \mathbf{B} \in \mathbb{R}^{n_1 \times n_2}$, $\mathbf{Y} = \mathbf{A} \odot \mathbf{B}$ denotes the Hadamard product of $\mathbf{A}$ and $\mathbf{B}$, such that $Y_{ij} = A_{ij} B_{ij}$, for all $i, j$. Sets are represented by Calligraphic letters, like $\mathcal{A}, \mathcal{B}$. For a finite set $\mathcal{A}$, $|\mathcal{A}|$ denotes the number of elements in $\mathcal{A}$. Throughout the paper, $\log$ refers to the logarithm in base 2, while $\ln$ denotes the natural logarithm.

## 2 Related Work

**UNNs for SCI.** While the majority of SCI recovery algorithms developed for various applications fall under classic optimization-based methods (e.g., [8, 9, 10]) or supervised DNN-based methods [24], in recent years, there has been increasing interest in leveraging UNNs in solving inverse problems. In SCI recovery, this trend has been motivated by the diversity of applications and datasets encountered

in various SCI applications, necessitating the availability of pre-trained denoising networks tailored to different resolutions and noise levels for various datasets. Another challenge with these traditional solutions is their robustness to various problem settings, such as measurement noise. These challenges have spurred a notable interest in developing solutions that harness the ability of DNNs to capture complex source models while not relying on training data.

While deep image priors (DIPs) have been applied to various inverse problems [29, 30, 31], their application to SCI recovery has been limited. The authors in [27] developed an iterative DIP-based solution for hyperspectral SCI. To enhance the performance and address the challenges faced by DIP-based methods in terms of falling into local minimas, they initialize the algorithm by the solutions obtained by GAP-TV [9]. In [26], the authors propose Factorized Deep Video Prior (DVP), which is a DIP-based SCI recovery algorithm for videos, which is based on separating the video into foreground and background and treating them separately. [32] develops a DIP-based solution for compressed ultrafast photography (CUP), where in addition to the normal SCI 2D measurement and additional side information consisting of the integral of all the frames (referred to as the time-unsheared view in [32]) is also collected. The video is reconstructed using an end-to-end approach using the DIP to enforce the source model. In [33], the authors leverage the concept of video snapshot compressive imaging (SCI) reconstruction to develop an algorithm for snapshot temporal compressive microscopy. They propose an iterative algorithm that utilizes UNNs to incorporate the source structure.

In the context of image recovery from underdetermined measurements corrupted by speckle noise, the authors in [28] recently proposed the idea of bagged-DIP, which is based on independently training multiple DIPs operating at different frame sizes and averaging the results. In this paper, we extend the idea to videos and construct a bagged-DVP, which as we show in our experimental results robustly achieves state-of-the-art performance among all UNN-based SCI video recovery methods.

**Mask optimization.** In various SCI applications, one can design the masks, which are typically binary-valued, and used for modulating the input 3D data cube. This naturally raises the question of optimizing the masks to improve the performance. To address this problem, several empirical works have designed solutions that simultaneously solve the SCI recovery problem and optimize the masks. In [34], the authors design an end-to-end autoencoder network to train the reconstruction and mask simultaneously for video data and find the trained mask has some distribution such as non-zero probability around 0.4 and varies smooth spatially and temporally. Similarly, in [35], deep unfolding style networks are trained to simultaneously reconstruct 3D images and also optimize the binary masks. They show that for the *empirically* jointly optimized masks have a non-zero probability of around 0.4. The authors in [16] design an end-to-end VIT-based SCI video recovery solution that simultaneously learns the reconstruction signal and the mask. They consider a special type of mask that constrained by their hardware design.

Due to the highly non-convex nature of the described joint optimization problem, empirically-jointly-optimized solutions are likely to converge to suboptimal results. Furthermore, the optimized solution, inherently dependent on the training data, lacks theoretical guarantees. To address these limitations, [36] employed a compression-based framework to theoretically optimize the binary-valued masks in the case of noiseless measurements and showed that in that case the optimized probability of non-zero entries is always smaller than 0.5. Here, we theoretically characterize the performance of UNN-based SCI recovery methods and show a consistent result in the case of noise-free measurements. Interestingly, as shown in our experiments, for noisy measurements, the optimized probability can be larger than 0.5. We derive novel theoretical results explaining this phenomenon.

## 3 DIP for SCI inverse problem

### 3.1 SCI inverse problem

The objective of a SCI system is to reconstruct a three-dimensional (3D) data cube from its two-dimensional (2D) compressed measurement. Specifically, let $\mathbf{X} \in \mathbb{R}^{n_1 \times n_2 \times B}$ represent the target 3D data cube. In an SCI system, $\mathbf{X}$ is mapped to a singular measurement frame $\mathbf{Y} \in \mathbb{R}^{n_1 \times n_2}$. This mapping, particularly as implemented in hyperspectral SCI and video SCI [2], can be modeled as a linear system as follows [2, 37]: $\mathbf{Y} = \sum_{i=1}^{B} \mathbf{C}_i \odot \mathbf{X}_i + \mathbf{Z}$. Here, $\mathbf{C} \in \mathbb{R}^{n_1 \times n_2 \times B}$ represents the sensing kernel (or mask), and $\mathbf{Z} \in \mathbb{R}^{n_1 \times n_2}$ denotes the additive noise. The terms $\mathbf{C}_i = \mathbf{C}(:, :, i)$ and

$\mathbf{X}_i = \mathbf{X}(:,:,i) \in \mathbb{R}^{n_1 \times n_2}$ correspond to the $b$-th sensing kernel (mask) and the associated signal frame, respectively.

To simplify the mathematical representation of the system, we vectorize each frame as $\mathbf{x}_i = \mathrm{Vec}(\mathbf{X}_i) \in \mathbb{R}^n$ with $n = n_1 n_2$. Then, we vectorize the data cube $\mathbf{X}$ by concatenating the $B$ vectorized frames into a column vector $\mathbf{x} \in \mathbb{R}^{nB}$ as $\mathbf{x} = \left[\mathbf{x}_1^\top, \ldots, \mathbf{x}_B^\top\right]^\top$. Similarly, we define $\mathbf{y} = \mathrm{Vec}(\mathbf{Y}) \in \mathbb{R}^n$ and $\mathbf{z} = \mathrm{Vec}(\mathbf{u}) \in \mathbb{R}^n$. Using these definitions, the measurement process can also be expressed as

$$\mathbf{y} = \mathbf{H}\mathbf{x} + \mathbf{z}. \tag{1}$$

The sensing matrix $\mathbf{H} \in \mathbb{R}^{n \times nB}$, is a highly sparse matrix that is formed by the concatenation of $B$ diagonal matrices as

$$\mathbf{H} = [\mathbf{D}_1, ..., \mathbf{D}_B], \tag{2}$$

where, for $i = 1, \ldots B$, $\mathbf{D}_i = \mathrm{diag}(\mathrm{Vec}(\mathbf{C}_i)) \in \mathbb{R}^{n \times n}$. Using this notation, the measurement vector can be written as $\mathbf{y} = \sum_{i=1}^B \mathbf{D}_i \mathbf{x}_i$ The goal of a SCI recovery algorithm is to recover the data cube $\mathbf{x}$ from undersampled measurements $\mathbf{y}$, while having access to the sensing matrix (or mask) $\mathbf{H}$.

## 3.2 Theoretical analysis of DIP-based SCI recovery

The Deep Image Prior (DIP) [25] hypothesis provides a framework for understanding the potential of UNNs in capturing the essence of complex source structures without requiring training data. Define $\mathcal{Q} \subseteq \mathbb{R}^n$ as the class of signals of interest (e.g., class of video signals consisting of $B$ frames.). Also, let $g_\theta : \mathbb{R}^p \to \mathbb{R}^n$ represent a UNN parameterized by $\theta \in \mathbb{R}^k$. Informally, DIP hypothesis states that any signal in $\mathcal{Q}$ can be presented as the output of the DIP parameterized by parameters $\theta \in \mathbb{R}^k$. This can be represented more formally as follows.

**DIP hypothesis:** Assume that $\mathbf{u} \in \mathbb{R}^p$ is sampled i.i.d. from a uniform distribution $\mathcal{U}(0,1)$. For any $\mathbf{x} \in \mathcal{Q}$, the DIP hypothesis states that for any $\mathbf{x} \in \mathbb{R}^k$, there exists $\theta \in [0,1]^k$, such that $\|g_\theta(\mathbf{u}) - \mathbf{x}\|_2 \le \delta$, almost surely.

This hypothesis underscores the capability of UNNs to function as powerful priors, capturing intricate data structures inherent in natural images and other complex datasets, thereby bridging the gap between classical analytic methods and modern machine learning techniques.

Given SCI measurements $\mathbf{y} = \mathbf{H}\mathbf{x} + \mathbf{z}$, as described in (1) with $\mathbf{H}$ defined in (2), a DIP represented by $g_\theta : \mathbb{R}^p \to \mathbb{R}^n$ can be used to recover $\mathbf{x}$ from measurements $\mathbf{y}$ as follows: Step 1) Randomly sample $\mathbf{u}$ (independent of $\mathbf{y}$ and $\mathbf{H}$), as required by the DIP. Step 2) Solve the DIP-SCI optimization:

$$\hat{\mathbf{x}} = \arg\min \|\mathbf{y} - \mathbf{H}\mathbf{c}\|_2, \quad \text{subject to } \mathbf{c} = g_\theta(\mathbf{u}), \ \theta \in [0,1]^k. \tag{3}$$

Before describing our proposed approach to solving DIP-SCI optimization in Section 4, we theoretically characterize the performance of (3), under noise-free and noisy measurements and use our theoretical results to i) bound the number of frames that can be recovered from a single 2D measurement, and ii) optimize the parameters of the mask $\mathbf{H}$ that is used for modulating the data.

### 3.2.1 Noise-free measurements

The following theorem characterizes the performance of (3) in case where the measurements are noise-free and connects its performance ($\|\mathbf{x} - \hat{\mathbf{x}}\|_2$) to the ambient dimension $n$, number of frames $B$, number of parameters of the DIP $k$, the distortion $\delta$ and the Lipschitz coefficient $L$.

**Theorem 3.1.** *Let $\mathbf{x} \in \mathcal{Q}$, where $\mathcal{Q}$ denotes a compact subset of $\mathbb{R}^n$, such that $\|\mathbf{x}\|_\infty \le \frac{\rho}{2}$, for all $\mathbf{x} \in \mathcal{Q}$. Assume that $g_\theta(\mathbf{u}) : [0,1]^N \to \mathbb{R}^{nB}$ is $L$-Lipschitz as a function of $\theta$. Let $\mathbf{y} = \mathbf{H}\mathbf{x}$, where $\mathbf{H} = [\mathbf{D}_1, \ldots, \mathbf{D}_B]$, where $\mathbf{D}_i = \mathrm{diag}(D_{i,1}, \ldots, D_{i,n})$, $i = 1, \ldots, B$, are independently generated with $D_{i,1}, \ldots, D_{i,n}$ i.i.d. $\mathrm{Bern}(p)$. Given randomly generated $\mathbf{u}$, let $\hat{\mathbf{x}}$ denote the solution of (3). Then, if $\min_{\mathbf{c}: \ \mathbf{c}=g_\theta(\mathbf{u}), \theta \in [0,1]^k} \frac{1}{nB}\|\mathbf{x} - \mathbf{c}\|_2 \le \delta$, we have*

$$\frac{1}{\sqrt{nB}}\|\mathbf{x} - \hat{\mathbf{x}}\|_2 \le \sqrt{1 + \frac{Bp}{1-p}}\delta + \frac{2\rho}{\sqrt{p(1-p)}}\left(\frac{kB^2 \log\log n}{n}\right)^{\frac{1}{4}} + \frac{L}{\log n}\sqrt{\frac{k}{nB}}\left(\frac{B}{\sqrt{p(1-p)}} + 1\right), \tag{4}$$

*with a probability larger than $1 - 2^{-0.5k \log\log n + 1}$.*

The bound in (4) consists of multiple terms. The first term, i.e., $\sqrt{1 + \frac{Bp}{1-p}}\delta$, accounts for the effect of the DIP representation error. For instance if $\mathbf{x}$ is directly selected from the output space of DIP, then $\delta = 0$. The goal of the following two corollaries to shed light on the interplay of the three terms in (4) and highlight their implications on the performance of DIP-SCI optimization. First, Corollary 3.2 characterizes an upper bound on the number of frames $B$ that are to be recovered from a single 2D measurement.

**Corollary 3.2.** *Consider the same setup as in Theorem 3.1. If*

$$B \leq \sqrt{\frac{n}{k(\log n)(\log \log n)}}, \tag{5}$$

*then* $\frac{1}{\sqrt{n}}\|\mathbf{x} - \hat{\mathbf{x}}\|_2 \leq \sqrt{1 + \frac{Bp}{1-p}}\delta + \frac{c_n}{\sqrt{p(1-p)}}$, *where* $c_n = O(1/(\log n)^{\frac{1}{4}})$ *does not depend on* $p$.

Next, Corollary 3.3 states that in the case where the measurements are not corrupted by noise, the value of $p$, the probability of a mask entry being non-zero, that minimizes the upper bound in (4) is always less than $0.5$. This is consistent with the results established i) empirically in the literature [16] and ii) theoretically in [36] using a compression-based framework.

**Corollary 3.3.** *Consider the same setup as in Theorem 3.1. The upper bound in (4) is minimized at* $p^* \in (0, 0.5)$.

### 3.2.2 Noisy measurements

In many practical SCI applications, the measurements are corrupted by additive noise. This raises the following natural question: How does the inclusion of noise in the model affects the optimized mask parameters? To address this question, we develop two theoretical results: Theorem 3.4 characterizing the reconstruction error $\|\mathbf{x} - \hat{\mathbf{x}}\|_2$ and Theorem 3.5 bounding the error in estimating the mean of the input frames $\bar{\mathbf{x}} = \frac{1}{B}\sum_{i=1}^{B}\mathbf{x}_i$. As we explain later, the combination of these two results provide a theoretical understanding on the performance of SCI recovery methods in the presence of noise and the corresponding optimized masks.

**Theorem 3.4.** *Consider the same setup as in Theorem 3.1. For* $\mathbf{x} \in \mathcal{Q}$, *let* $\mathbf{y} = \sum_{i=1}^{B}\mathbf{D}_i\mathbf{x}_i + \mathbf{z}$, *where* $\mathbf{z} \in \mathbb{R}^n$ *denotes the additive noise and* $\mathbf{z} \sim \mathcal{N}(\mathbf{0}, \sigma_z^2 I_n)$, *for some* $\sigma_z \geq 0$. *Let* $\hat{\mathbf{x}}$ *denote the solution of DIP-SCI optimization* (3). *If* $B$ *satisfies the bound in* (5), *then*

$$\frac{1}{\sqrt{nB}}\|\mathbf{x} - \hat{\mathbf{x}}\|_2 \leq \delta\sqrt{1 + \frac{Bp}{1-p}} + \frac{3\sigma_z}{p(1-p)}\sqrt{\frac{1}{\log n}}$$
$$+ \left(\frac{8}{\log n}\right)^{\frac{1}{4}}\sqrt{\frac{\delta\sigma_z}{p(1-p)}}(1 + \alpha_n) + \sqrt{\frac{1}{p(1-p)}}\frac{\rho}{(\log n)^{\frac{1}{8}}}(1 + \beta_n) + \gamma_n, \tag{6}$$

*with a probability larger than* $1 - (2^{-0.5k\log\log n + 3} + e^{-0.3n})$. *Here,* $\alpha_n = O(\frac{1}{\sqrt{\log n}})$, $\beta_n = o(\frac{1}{(\log n)^{\frac{1}{4}}})$ *and* $\gamma_n = o(\frac{1}{\log n})$ *do not depend on* $\sigma_z$ *and* $p$.

**Theorem 3.5.** *Consider the same setup as in Theorem 3.4. Assuming that* $B$ *satisfies the bound in* (5), *then with probability larger than* $1 - (2^{-0.5k\log\log n + 3} + e^{-0.3n})$,

$$\frac{1}{\sqrt{n}}\|\frac{1}{B}\sum_{i=1}^{B}(\mathbf{x}_i - \hat{\mathbf{x}}_i)\|_2 \leq \delta\sqrt{1 + \frac{1}{pB}} + \frac{1}{p}\sqrt{\frac{2\rho\sigma_z}{B}}\left(\frac{k\log\log n}{n}h(p)\right)^{\frac{1}{4}} + \frac{1}{p\sqrt{B}}\upsilon_n + \frac{L}{\log n}\sqrt{\frac{k}{nB}},$$

*where* $\upsilon_n = O((\log n)^{-\frac{1}{8}})$ *and does not depend on* $p$.

To shed light on the implications of these two theorems, the following corollary characterizes the value of $p$ optimizing each bound.

**Corollary 3.6.** *Consider the same setting as Theorem 3.4. The upper bound in Theorem 3.4 is always optimized at* $p^* < 0.5$. *On the other hand, the upper bound in Theorem 3.5 is a decreasing function of* $p$ *and is minimized at* $p^* = 1$.

Let $\bar{\mathbf{x}}_B = [\bar{\mathbf{x}}^\top, \ldots, \bar{\mathbf{x}}^\top]^\top \in \mathbb{R}^{nB}$, i.e., the reconstruction signal derived by repeating the average frame $\bar{\mathbf{x}} = \frac{1}{B}\sum_{i=1}^{B}\mathbf{x}_i$. Then, using the triangle inequality, we have

$$\|\mathbf{x} - \hat{\mathbf{x}}\|_2 \leq \|\mathbf{x} - \bar{\mathbf{x}}_B\|_2 + \|\bar{\mathbf{x}}_B - \hat{\mathbf{x}}\|_2.$$

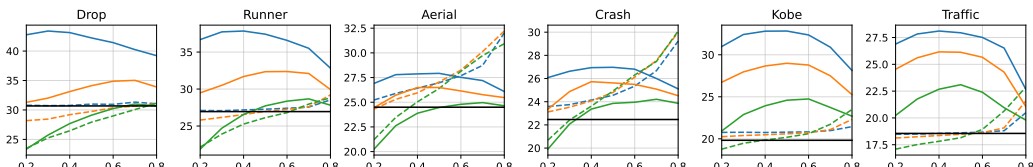

Figure 1: PSNR, shown as y-axis, of $\|\mathbf{x} - \hat{\mathbf{x}}\|_2$, $\|\hat{\mathbf{x}} - \bar{\mathbf{x}}_B\|_2$ and $\|\mathbf{x} - \bar{\mathbf{x}}_B\|_2$: masks are generated as Bern($p$), $p$ shown as x-axis,. Blue, orange and green lines represent noise levels of $\sigma = 0$, 10 and 25, respectively. Solid black line shows $\|\mathbf{x} - \bar{\mathbf{x}}_B\|_2$. Solid colored lines and dashed colored lines represent $\|\mathbf{x} - \hat{\mathbf{x}}\|_2$ and $\|\hat{\mathbf{x}} - \bar{\mathbf{x}}_B\|_2$, respectively.

Figure 1 shows $\|\mathbf{x} - \hat{\mathbf{x}}\|_2$, $\|\mathbf{x} - \bar{\mathbf{x}}_B\|_2$, and $\|\bar{\mathbf{x}}_B - \hat{\mathbf{x}}\|_2$, for different video test samples. Here are our key observations: 1) $\|\bar{\mathbf{x}}_B - \hat{\mathbf{x}}\|_2$ is an increasing function of $p$, which is consistent with Corollary 3.6. 2) The optimal value of $p^*$ that minimizes $\|\mathbf{x} - \hat{\mathbf{x}}\|_2$, is an increasing function of $\sigma_z$, for all test videos. 2) In cases where the difference between $\bar{\mathbf{x}}_B$ and $\mathbf{x}$ is relatively large, e.g. Traffic, the optimized $p^*$ stays smaller than 0.5, even for large values of $\sigma_z$, as predicted by Theorem 3.4. 3) On the other hand, in cases where $\bar{\mathbf{x}}_B$ provides a high-fidelity representation of $\mathbf{x}$ and $\|\bar{\mathbf{x}}_B - \mathbf{x}\|_2$ is relatively small (e.g., Drop), for large values of noise power, the optimal value of $p^*$ can move beyond 0.5, as predicted by Theorem 3.5. In other words, in such cases, the algorithm moves toward estimating the mean of the frames, which is a good representation of the actual data frame.

## 4    SCI-BDVP: Bagged-DVP for video SCI

Recall the DIP-SCI optimization described in (3), i.e., $\hat{\mathbf{x}} = \arg\min_{\mathbf{c}} \|\mathbf{y} - \mathbf{Hc}\|_2$, where $\mathbf{c} = g_\theta(\mathbf{u})$, $\theta \in [0,1]^k$ and $\mathbf{u}$ generated independently and randomly according to a pre-specified distribution. To solve this optimization, one straightforward approach is to solve $\min_\theta f(\theta)$, with $f(\theta) = \|\mathbf{y} - \mathbf{H}g_\theta(\mathbf{u})\|_2^2$, by directly applying gradient descent to the differentiable function $f(\theta)$. However, given the highly non-linearity and non-convexity of $f(\theta)$, this approach is prone to readily getting trapped into local a minima and achieving considerably sub-optimal performance. Generally, a better approach to is to write the DIP-SCI optimization as $\hat{\mathbf{x}} = \arg\min_{\mathbf{c} \in \mathcal{C}(\mathbf{u})} \|\mathbf{y} - \mathbf{Hc}\|_2^2$, where $\mathcal{C}(\mathbf{u}) \triangleq \{\mathbf{c} = g_\theta(\mathbf{u}) : \theta \in [0,1]^k\}$. This alternative presentation of the problems leads to minimizing a convex cost function over a non-convex set. A classic approach to solve this optimization is projected gradient descent (PGD), which while in general is not guaranteed to converge to the global minima is more apt to recover a solution in the vicinity of the desired signal.

**Remark 4.1.** *Theoretical feasibility of SCI recovery was first established in [38] using a compression-based framework for modeling source structure. There, the authors considered $\hat{\mathbf{x}} = \arg\min_{\mathbf{c} \in \mathcal{C}} \|\mathbf{y} - \mathbf{Hc}\|_2^2$, where $\mathcal{C}$ denotes a* discrete *set of the codewords of a compression code. They theoretically proved that in that case, despite the non-convexity of the problem, PGD is able to converge to the vicinity of the desired signal.*

The PGD applied to $\hat{\mathbf{x}} = \arg\min_{\mathbf{c} \in \mathcal{C}(\mathbf{u})} \|\mathbf{y} - \mathbf{Hc}\|_2^2$ proceeds as follows: Start form an initialization point $\mathbf{x}_0$. For $t = 1, 2, \ldots, T$, perform the following two steps i) Gradient descent: $\mathbf{x}_{t+1}^G = \mathbf{x}_t + \mu\mathbf{H}^\top(\mathbf{y} - \mathbf{Hx}_t)$, and ii) Projection: $\mathbf{x}_{t+1} = \arg\min_{\mathbf{c} \in \mathcal{C}(\mathbf{u})} \|\mathbf{c} - \mathbf{s}_{t+1}\|_2$, or

$$\hat{\theta}_{t+1} = \arg\min_\theta \|g_\theta(\mathbf{u}) - \mathbf{x}_{t+1}^G\|_2, \quad \mathbf{x}_{t+1} = g_{\hat{\theta}_{t+1}}(\mathbf{u}) \qquad (7)$$

To solve the non-convex optimization required at the projection step, one can again employ gradient descent. However, in addition to the non-convexity of the cost function, another common known issue with projection into the domain of a DIP is overfitting [25, 6, 39, 40]. Moreover, in PGD, ideally one needs to set the resolution of the projection step adaptively, such that during the initial steps the DIP has a coarser resolution and as it proceeds it becomes finer and finer. This poses the following question: Which DIP structure should one use to optimize the final performance?

To address this question, the authors in [28] have proposed, bagged-DIP, which consists of employing multiple DIP with different structures in parallel, for the DIP projection step and averaging the outputs. They show that this approach provides a robust projection module which consistently outperforms the performance achievable by each individual DIP network, and also provides, at least partially, the flexibility and adaptability required by PGD.

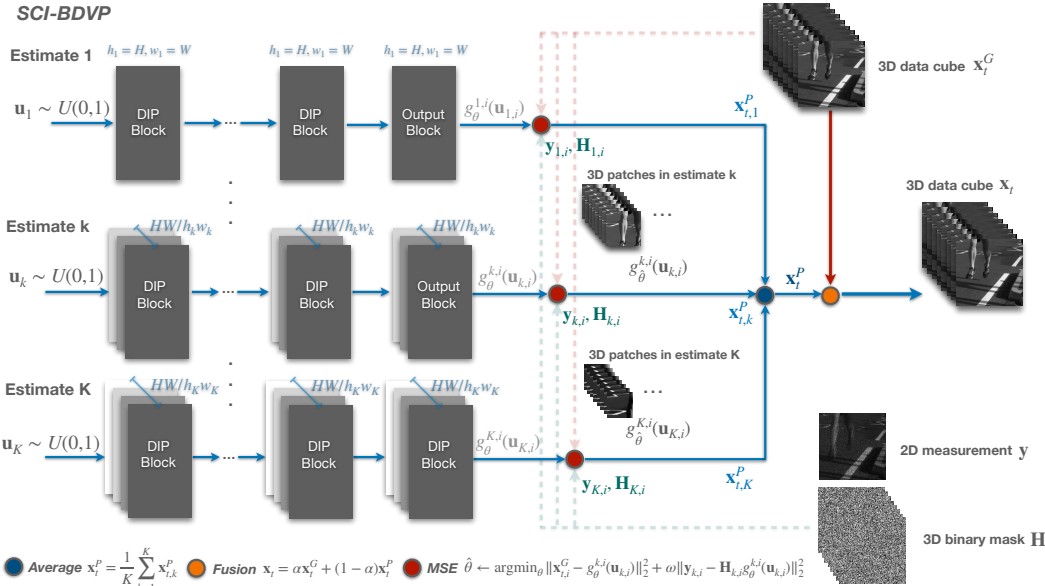

Figure 2: The structure of SCI-BDVP. There are $K$ estimates generated, each using a different patch size. The blue dot denotes averaging the $K$ estimates, the orange dot denotes averaging $\mathbf{x}_t^P$ and $\mathbf{x}_t^G$ with weight $\alpha$, the red dot denotes the loss function used for training the DIP parameters, requiring $\mathbf{x}_t^G$, $\mathbf{y}$ and $\mathbf{H}$. The red lines denote using 3D gradient descent result $\mathbf{x}_t^G$, which is used for training the parameters of the $k$-th DIP (red dot), and averaging with projection output $\mathbf{x}_t^P$ (orange dot). The green lines denote using 2D measurement y and 3D binary mask H for training parameters of DIPs in different estimate $k$.

Bagged-DIP, essentially employs bagging idea to mitigate overfitting. As the DIP projection iterations proceeds (within each step of PGD), overfitting tends to occur after a certain threshold. However, due to the variance reduction facilitated by bagging, the bagged estimate can demonstrate less overfitting. In other words, the bagged estimate is less sensitive to the stopping time of the DIP training. In essence, each DIP is not required to produce the best estimate at every iteration of PGD.

Inspired by the bagged-DIP solution, here we propose the bagged-DVP for SCI (SCI-BDVP), as shown in Figure 3. SCI-BDVP, in addition to the standard gradient descent (GD) step, defined as $\mathbf{x}_{t+1}^G = \mathbf{x}_t + \mu\mathbf{H}^T(\mathbf{y} - \mathbf{H}\mathbf{x}_t)$, consists of two main additional components: i) The bagged-DVP module that simultaneously projects the output of the GD step onto the domain of multiple DVP networks operating at varying patch sizes (refer to Figure 2 and then averages their outputs, and ii) a skip connection that computes a weighted average of the output of the GD step and the bagged-DVP step. Next, we briefly explain the detailed construction of each component.

**SCI-BDVP.** Figure 2 schematically shows the structure of a bagged-DVP consisting of $K$ individual DVPs, each operating at a different scale and trained separately. More specifically, for each $k$, $k = 1, \cdots, K$, the $3D$ video is partitioned into non-overlapping video cubes of dimensions $(h_k, w_k)$. For each video cube of dimension $(h_k, w_k, B)$, we train a separate DVP. In other words, at scale $k$, we need to train $N_k = H/h_k \times W/w_k$ separate DVPs. (The total aggregate number of DVPs that are trained is going to be $\sum_{k=1}^K N_k$.) At each scale, the separately projected video cubes are concatenated to form $\mathbf{x}_{t+1,k}^P$, a video frame of the same dimensions as the desired video. At scale $k$, let $g_\theta^k(\cdot)$ denote a DVP that generates an output video frame of dimensions $h_k \times w_k \times B$. To cover the whole video frame at scale $k$, we need to train $N_k$ separate DVPs $g_\theta^{k,i}(\cdot)$, each having an independently drawn input, $\mathbf{u}_{k,i}$. $i$ denotes the index of partitioned video cube. To train each of these $N_k$ DVPs, we first extract the corresponding parts from $\mathbf{x}_{t+1}^G$, $\mathbf{y}$ and $\mathbf{H}$ and denote them as $\mathbf{x}_{t+1,i}^G$, $\mathbf{y}_i$ and $\mathbf{H}_i$, respectively.[2] Then, to train the corresponding DVP to form reconstruction $\mathbf{x}_{t+1,k,i}^P$, we minimize

---

[2]Note that given the special structure of the sensing matrix $\mathbf{H}$ in SCI, given a part of the input video frame of the same depth $B$, one can readily extract the corresponding mask portion and measurements.

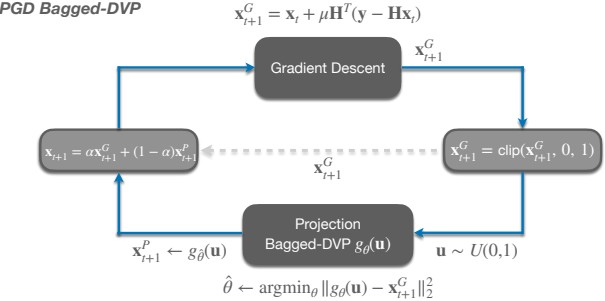

Figure 3: SCI-BDVP (GD): Iterative PGD-type algorithm. Each step consists of GD and BDVP projection, with an additional skip-connection.

$\|\mathbf{x}_{t+1,i}^G - g_\theta^{k,i}(\mathbf{u}_{k,i})\|_2^2 + \omega\|\mathbf{y}_i - \mathbf{H}_i g_\theta^{k,i}(\mathbf{u}_{k,i})\|_2^2$, where $\omega > 0$ denotes the regularization parameter. Unlike classic DIP cost function, here we use the measurements $\mathbf{y}$ as an additional regularizer. After recovering $\mathbf{x}_{t+1,k,i}^P$, $i = 1, \ldots, N_k$, we concatenate them based on their locations to form $\mathbf{x}_{t+1,k}^P$. We repeat the same process, for each $k = 1, \ldots, K$ to find $\mathbf{x}_{t+1,1}^P, \ldots, \mathbf{x}_{t+1,K}^P$. Finally, we use the idea of bagging and define $\mathbf{x}_{t+1}^P = \frac{1}{K}\sum_{k=1}^K \mathbf{x}_{t+1,k}^P$.

**Skip connection.** After obtaining $\mathbf{x}_{t+1}^P$ and $\mathbf{x}_{t+1}^G$, we define $\mathbf{x}_{t+1}$ as their weighted average: $\mathbf{x}_{t+1} = \alpha\mathbf{x}_{t+1}^G + (1-\alpha)\mathbf{x}_{t+1}^P$, where $\alpha \in (0,1)$. (See Figure 3.) In the experiments in Appendix C.3, we show how the addition of this skip connection consistently improves the achievable performance.

## 5 Experiments

We evaluate the performance of SCI-BDVP and compare it with existing SCI methods, for $\sigma_z = 0$ and $\sigma_z > 0$. Our experimental results are consistent with our theoretical results on mask optimization. To evaluate the performance we use peak-signal-to-noise-ratio (PSNR) and structured similarity index metrics (SSIM) [41]. All the tests are performed on a single NVIDIA RTX 4090 GPU.

**Datasets and baselines.** We compare our method against the baselines on 6 gray-scaled benchmark videos including `Kobe`, `Runner`, `Drop`, `Traffic`, `Aerial`, `Vehicle` [18], where the spatial resolution is $256 \times 256$, and $B = 8$. We choose 5 representative baseline methods i) GAP-TV [9] : the Plug-and-play (PnP) method that employs a total-variation denoiser; ii) PnP-FFDnet [17] and PnP-FastDVDnet [18] : PnP methods that employ pre-trained deep denoisers, iii) PnP-DIP [27]: DIP-based iterative method; iv) Factorized-DVP [26]: Untrained End-to-End (E2E) network. Baseline setups follows that exactly stated in the respective papers. The details of proposed SCI-BDVP can be found in Appendix B.

**Masks for noiseless and noisy measurements.** For the case of SCI without noise, we obtain the measurements from equation (1), where we randomly sample mask values from $\text{Bern}(p)$ with $p = 0.2, 0.3, \ldots, 0.8$. For the noisy setup, zero-mean Gaussian noise with variance $(\sigma^2)$, $\sigma = 10$, $\sigma = 25$ and $\sigma = 50$, is added to the measurements. For the results reported in Tables 1 and 2, the masks are randomly and independently generated as as $\text{Bern}(0.5)$.

### 5.1 Reconstruction results for video SCI

**Noiseless measurement.** In Table 1 we compare the performance of SCI-BDVP against baselines. To highlight the effectiveness of the bagged DVP idea, we also implemented two versions of our proposed method: i) SCI-BDVP (E2E), an end-to-end BDVP-based solution and 2) SCI-BDVP (GAP): an iterative algorithm that employs generalized alternative projection (GAP) update rule and BDVP projection (Refer to Appendix B.1 for a description of GAP and GD and our rationale for the choice of each method.). It can be observed that both SCI-BDVP (E2E) and SCI-BDVP (GAP) outperform existing untrained methods. Specifically, SCI-BDVP (GAP) achieves state-of-the-art performance and on average improves about 1 dB in PSNR compared to other methods.

**Noisy measurement.** Table 2 compares the performance of SCI-BDVP (E2E) and SCI-BDVP (GD) with baseline methods. As explained in Appendix B.1, unlike noise-free measurements, in the case of noisy measurements, especially when noise variance grows, GAP update rule is no longer a reasonable

Table 1: **Reconstruction results on Noise-free measurements.** PSNR (dB) (left entry) and SSIM (right entry) of different algorithms. Best results are in **bold**, second-best results are underlined.

| Dataset | Kobe | Traffic | Runner | Drop | Crash | Aerial | Average |
|---|---|---|---|---|---|---|---|
| GAP-TV | 22.38, 0.666 | 19.60, 0.609 | 28.15, 0.884 | 32.49, 0.949 | 24.46, 0.842 | 25.65, 0.835 | 25.46, 0.798 |
| PnP-FFD | 30.39, 0.924 | 23.89, 0.830 | 32.66, 0.935 | 39.82, 0.986 | 24.18, 0.819 | 24.57, 0.836 | 25.46, 0.798 |
| PnP-FastDVD | **32.79**, **0.948** | **27.89**, **0.929** | **37.52**, **0.967** | **42.35**, **0.989** | **26.76**, **0.921** | **27.92**, **0.897** | **32.54**, **0.942** |
| PnP-DIP | 22.52, 0.627 | 20.27, 0.617 | 29.54, 0.878 | 31.23, 0.908 | 24.33, 0.751 | 25.45, 0.790 | 25.56, 0.762 |
| Factorized-DVP | 25.54, 0.740 | 23.38, 0.760 | 30.76, 0.890 | 36.69, 0.970 | 26.05, 0.850 | 26.84, 0.860 | 28.21, 0.845 |
| **SCI-DVP (E2E)** | 25.24, 0.741 | 18.89, 0.503 | 26.92, 0.852 | 35.00, 0.958 | 21.82, 0.653 | 21.31, 0.684 | 24.87, 0.732 |
| **SCI-BDVP (E2E)** | 27.76, 0.866 | 22.00, 0.741 | 32.86, 0.939 | 39.67, 0.985 | 23.59, 0.805 | 23.98, 0.809 | 28.31, 0.857 |
| **SCI-BDVP (GAP)** | 28.42, 0.886 | 22.84, 0.779 | 34.32, 0.954 | 40.76, 0.986 | 24.96, 0.851 | 25.16, 0.837 | 29.41, 0.882 |

Table 2: **Reconstruction Results on Noisy Measurements.** PSNR (dB) (left entry) and SSIM (right entry) of different algorithms. Best results are highlighted in **bold**, second-best results are underlined.

| Dataset | $\sigma$ | Explicit Regularizor | Learning-based supervised methods | | | Learning-based unsupervised methods | | |
|---|---|---|---|---|---|---|---|---|
| | | GAP-TV | FFD | FastDVD (GAP) | FastDVD (PGD) | PnP-DIP | **SCI-BDVP (E2E)** | **SCI-BDVP (GD)** |
| Kobe | 10 | 22.16, 0.580 | 25.68, 0.706 | **28.94**, **0.811** | 22.96, 0.595 | 22.47, 0.562 | 25.76, 0.741 | 26.39, 0.805 |
| | 25 | 21.65, 0.461 | 21.37, 0.436 | 24.44, 0.564 | 22.62, 0.606 | 21.34, 0.404 | 22.24, 0.511 | **25.89**, **0.775** |
| | 50 | 20.29, 0.297 | 16.04, 0.188 | 19.59, 0.241 | 20.87, 0.524 | 19.52, 0.238 | 16.99, 0.242 | **23.34**, **0.640** |
| Traffic | 10 | 19.50, 0.565 | 20.56, 0.684 | **26.11**, **0.855** | 22.65, 0.769 | 19.95, 0.562 | 21.06, 0.649 | 22.66, 0.740 |
| | 25 | 19.23, 0.498 | 18.23, 0.524 | **22.77**, 0.692 | 21.64, 0.740 | 19.24, 0.464 | 18.94, 0.484 | 22.23, 0.718 |
| | 50 | 18.42, 0.385 | 13.90, 0.310 | 18.00, 0.367 | 18.76, 0.552 | 17.97, 0.344 | 15.14, 0.295 | **20.56**, **0.611** |
| Runner | 10 | 27.40, 0.766 | 26.69, 0.739 | **32.21**, 0.845 | 27.92, 0.844 | 27.22, 0.663 | 27.85, 0.764 | 31.15, **0.916** |
| | 25 | 25.99, 0.610 | 22.18, 0.518 | 27.63, 0.650 | 28.33, 0.856 | 24.93, 0.497 | 21.93, 0.410 | **30.31**, **0.895** |
| | 50 | 23.14, 0.398 | 15.74, 0.280 | 22.31, 0.361 | **27.04**, **0.807** | 21.74, 0.322 | 16.31, 0.183 | 25.11, 0.693 |
| Drop | 10 | 30.75, 0.802 | 29.52, 0.765 | 33.81, 0.837 | 31.54, 0.932 | 29.12, 0.761 | 31.45, 0.870 | **35.03**, **0.962** |
| | 25 | 28.11, 0.614 | 23.36, 0.527 | 29.13, 0.646 | 32.52, 0.940 | 26.42, 0.842 | 23.46, 0.480 | **34.17**, **0.954** |
| | 50 | 24.09, 0.384 | 16.73, 0.298 | 23.40, 0.350 | 30.48, 0.856 | 26.46, 0.823 | 17.56, 0.233 | 29.86, **0.889** |
| Crash | 10 | 24.12, 0.728 | 21.83, 0.649 | **25.61**, 0.799 | 24.70, 0.790 | 23.46, 0.647 | 22.54, 0.655 | 25.57, **0.835** |
| | 25 | 23.40, 0.577 | 19.67, 0.458 | 24.09, 0.609 | 24.54, 0.795 | 22.11, 0.492 | 19.96, 0.379 | **25.33**, **0.821** |
| | 50 | 21.58, 0.376 | 15.33, 0.255 | 20.92, 0.342 | 23.35, 0.706 | 20.28, 0.294 | 15.79, 0.178 | **23.43**, 0.693 |
| Aerial | 10 | 25.21, 0.717 | 21.62, 0.641 | **26.51**, 0.763 | 23.07, 0.626 | 24.74, 0.671 | 22.84, 0.686 | 25.62, **0.817** |
| | 25 | 24.31, 0.570 | 19.66, 0.447 | 24.58, 0.586 | 23.99, 0.737 | 24.19, 0.613 | 20.39, 0.410 | **25.47**, **0.796** |
| | 50 | 22.16, 0.375 | 15.18, 0.237 | 21.49, 0.336 | 23.51, 0.713 | 21.07, 0.315 | 15.82, 0.179 | 22.97, 0.638 |
| Average | 10 | 25.21, 0.717 | 21.62, 0.641 | 26.51, 0.763 | 25.47, 0.760 | 24.49, 0.644 | 25.25, 0.727 | **27.73**, **0.846** |
| | 25 | 24.31, 0.570 | 19.66, 0.447 | 24.58, 0.586 | 25.61, 0.779 | 24.19, 0.613 | 21.15, 0.446 | **27.23**, **0.827** |
| | 50 | 22.16, 0.375 | 15.18, 0.237 | 21.49, 0.336 | 24.00, 0.693 | 21.17, 0.389 | 16.27, 0.218 | **24.21**, **0.694** |

choice. Therefore, for noisy data, we replace the GAP update rule with GD. For completeness, for PnP-FastDVDnet [18], we report both GAP-based version (as implemented in [18]) and GD-based version (newly implemented here). We observe SCI-BDVP (GD) considerably outperforms PnP-DIP [27][3]. Additionally, SCI-BDVP (GD) in most cases outperforms pre-trained method [18], across noise levels, while showing a robust performance on different datasets and noise levels.

**Time/computational complexity.** The proposed SCI-BDVP method relies on the bagging of multiple DIP projections. These DIP projections, which vary depending on the patch size, involve different levels of computational complexity. Table 3 shows the average time required to perform each DIP projection for each patch size. As observed, the time increases considerably as the patch size decreases. This is expected because the number of networks that need to be trained grows significantly. (Refer to Figure 2 for a pictorial representation.) Additional computational complexity analysis of our proposed method and its comparisons with other methods is included in Appendix B.3.

## 5.2 Mask optimization

We consider masks that are generated independent of the data as i.i.d.$\sim \mathrm{Bern}(p)$. The question is what value of $p$ optimizes the reconstruction performance? Figures 4 and 5 show the achieved reconstruction PSNR as a function of $p$, for the cases of noiseless and noisy measurements, respectively. For noiseless measurements, the results are shown both for SCI-BDVP (GAP) and PnP-FastDVDnet (GAP). It can be that for both methods, the optimized value of $p$ is smaller than 0.5 (around 0.4) and

---

[3]Since the code of Factorized-DVP [26] is not available online, we could only compare our results with the results reported for noise-free measurements.

Table 3: Time complexity of our proposed SCI-BDVP was evaluated on various patch sizes (64, 128, 256) of video blocks, using a standard 1000 DVP iterations for training.

| Patch size | # of patches | Time (min.) |
|---|---|---|
| 64 | 16 | 1.5 |
| 128 | 4 | 0.28 |
| 256 | 1 | 0.12 |

consistent with empirical observations reported in [35, 34]. For the noisy measurements, we see that $p^*$ is an increasing function of $\sigma_z$, consistent with our theoretical results discussed earlier in Section 3. (Refer to Appendix C.1 for further results.)

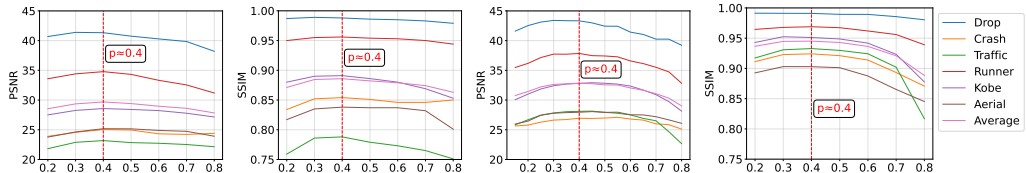

Figure 4: Reconstruction PSNR ($\|\mathbf{x} - \hat{\mathbf{x}}\|_2$) and SSIM as a function of $p$, using SCI-BDVP (GAP) (two leftmost figures) and PnP-FastDVDnet (GAP) (two rightmost figures). For each value of $p$, the masks are independently generated i.i.d.$\sim \mathrm{Bern}(p)$.

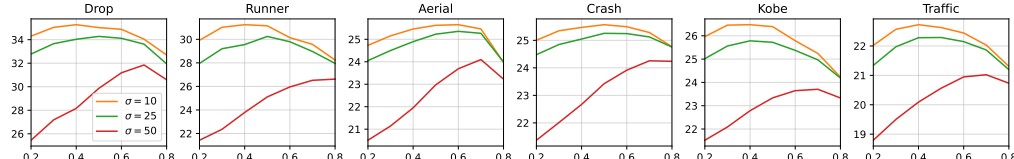

Figure 5: Reconstruction PSNR ($\|\mathbf{x} - \hat{\mathbf{x}}\|_2$) of SCI-BDVP (GD), y-axis, as a function of $p$, x-axis. For each value of $p$, the masks are independently generated i.i.d.$\sim \mathrm{Bern}(p)$.

## 6 Conclusion

We have studied application of UNNs SCI recovery. We propose an iterative solution with bagged DVP (multiple, separately trained DVPs with averaged outputs), achieving state-of-the-art performance among unsupervised solutions for noise-free measurements and robustly outperforming both supervised and UNN methods for noisy measurements. Additionally, we provide a theoretical framework analyzing the performance of UNN-based methods, characterizing achievable performance and guiding hardware parameter optimization. Simulations validate our theoretical findings.

An important application of SCI is hyperspectral snapshot imaging (HSI). Our results in this paper provide a theoretical foundation to understand HSI systems and optimize their hardware. Additionally, the developed theoretical framework can be used to explore aspects specific to HSI, such as masks being shifted versions of each other. We also expect our algorithm to effectively address overfitting in HSI tasks, enhancing reconstruction performance. We plan to explore these aspects further in our future research.

Several other aspects remain for future work. Theoretically, we only considered i.i.d. Bernoulli masks, while practical SCI systems typically are more constrained. Additionally, deriving information-theoretic lower bounds on SCI recovery is an open problem. Experimentally, we focused on classic baseline videos; exploring a richer set of samples and studying noise models beyond additive Gaussian noise are interesting directions for future research. We also plan to enhance the algorithm's efficiency by parallelizing the projections required by the bagged solution.

## Acknowledgements

MZ, XC and SJ were supported by NSF grant CCF-2237538.

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

# A Proofs

## A.1 Preliminary results and definitions

**Lemma A.1** (Concentration of $\chi^2$ [42]). *If $Z_1, Z_2, \ldots, Z_n$ are i.i.d. $\mathcal{N}(0, 1)$ random variables, then for any $t > 0$,*

$$P(\sum_{i=1}^{n} Z_i^2 \geq m(1+t)) \leq e^{-\frac{m}{2}(t-\log(1+t))}.$$

**Definition A.1.** *$f : \mathbb{R}^k \to \mathbb{R}$ is called an L-Lipschitz function (or L-Lipschitz continuous) if there exists a constant $L > 0$ such that for all $\mathbf{x}_1, \mathbf{x}_2 \in \mathbb{R}^k$:*

$$|f(\mathbf{x}_1) - f(\mathbf{x}_2)| \leq L||\mathbf{x}_1 - \mathbf{x}_2||_2.$$

*The constant $L$ is called the Lipschitz constant of $f$.*

## A.2 Proof of Theorem 3.1

Let $\tilde{\mathbf{x}} = \arg\min_{\mathbf{c}=g_\theta(\mathbf{u}):\ \theta\in\mathbb{R}^k} \|\mathbf{x} - \mathbf{c}\|_2$. By assumption $\frac{1}{\sqrt{nB}}\|\mathbf{x} - \tilde{\mathbf{x}}\|_2 \leq \delta$. On the other hand, since $\hat{\mathbf{x}} = \arg\min_{\mathbf{c}=g_\theta(\mathbf{u}):\ \theta\in\mathbb{R}^k} \| \mathbf{y} - \mathbf{Hc}\|_2^2$, $\|\mathbf{y} - \mathbf{H}\hat{\mathbf{x}}\|_2 \leq \|\mathbf{y} - \mathbf{H}\tilde{\mathbf{x}}\|_2$, where $\mathbf{y} = \mathbf{Hx}$. Therefore,

$$\|\mathbf{H}(\mathbf{x} - \hat{\mathbf{x}})\|_2 \leq \|\mathbf{H}(\mathbf{x} - \tilde{\mathbf{x}})\|_2 \tag{8}$$

Let $\hat{\mathbf{x}}_q = g_{[\hat{\theta}]_q}(\mathbf{u})$, i.e., the reconstruction corresponding to the $q$-bit quantized version of parameters $\theta$. By the triangle inequality,

$$\|\mathbf{H}(\mathbf{x} - \hat{\mathbf{x}})\|_2 = \|\mathbf{H}(\mathbf{x} - \hat{\mathbf{x}}_q + \hat{\mathbf{x}}_q - \hat{\mathbf{x}})\|_2 \geq \|\mathbf{H}(\mathbf{x} - \hat{\mathbf{x}}_q)\|_2 - \|\mathbf{H}(\hat{\mathbf{x}}_q - \hat{\mathbf{x}})\|_2. \tag{9}$$

Combining (8) with 9, it follows that

$$\|\mathbf{H}(\mathbf{x} - \hat{\mathbf{x}}_q)\|_2 \leq \|\mathbf{H}(\hat{\mathbf{x}}_q - \hat{\mathbf{x}})\|_2 + \|\mathbf{H}(\mathbf{x} - \tilde{\mathbf{x}})\|_2 \tag{10}$$

Given our assumption about the $L$-Lipschitz continuity of $g_\theta$ as a function of $\theta$, it follows that

$$\|\hat{\mathbf{x}}_q - \hat{\mathbf{x}}\|_2 = \|g_{[\hat{\theta}]_q}(\mathbf{u}) - g_{\hat{\theta}}(\mathbf{u})\|_2 \leq L\|\hat{\theta} - \theta\|_2 \leq L2^{-q}\sqrt{k}. \tag{11}$$

For $\mathbf{u} \in \mathbb{R}^{nB}$, using Cauchy-Schwartz inequality, $\|\mathbf{Hu}\|_2^2 = \sum_{j=1}^{n}(\sum_{i=1}^{B} D_{ij}u_{ij})^2 \leq \sum_{j=1}^{n}(\sum_{i=1}^{B} D_{ij}^2 \sum_{i=1}^{B} u_{ij}^2) \leq B(\max_{i,j} D_{ij}^2)\|\mathbf{u}\|_2^2 \leq B\|\mathbf{u}\|_2^2$, which follows since $D_{i,j} \in \{0, 1\}$. Therefore,

$$\|\mathbf{H}(\hat{\mathbf{x}}_q - \hat{\mathbf{x}})\|_2 \leq B\|\hat{\mathbf{x}}_q - \hat{\mathbf{x}}\|_2 \leq BL2^{-q}\sqrt{k}. \tag{12}$$

For a fixed random initialization $\mathbf{z} \in \mathbb{R}^p$, define the set of reconstructions derived from $q$-bit quantized parameters as $\mathcal{C}_q(\mathbf{u})$, i.e.,

$$\mathcal{C}_q(\mathbf{u}) = \{g_{[\theta]_q}(\mathbf{u}) :\ \theta \in [0, 1]^k\}.$$

Note that $|\mathcal{C}_q(\mathbf{u})| \leq 2^{qk}$. Given $\epsilon_1 > 0$, $\epsilon_2 > 0$, and $\mathbf{x}, \tilde{\mathbf{x}} \in \mathbb{R}^{nB}$, define events $\mathcal{E}_1$ and $\mathcal{E}_2$ as

$$\mathcal{E}_1 = \{\frac{1}{n}\|\mathbf{H}(\mathbf{x} - \tilde{\mathbf{x}})\|_2^2 \leq \frac{p^2}{n}\|\sum_{i=1}^{B}(\mathbf{x}_i - \tilde{\mathbf{x}}_i)\|_2^2 + \frac{p - p^2}{n}\|\mathbf{x} - \tilde{\mathbf{x}}\|_2^2 + B\rho^2\epsilon_1\}, \tag{13}$$

$$\mathcal{E}_2 = \{\frac{1}{n}\|\mathbf{H}(\mathbf{x} - \mathbf{c})\|_2^2 \geq \frac{p^2}{n}\|\sum_{i=1}^{B}(\mathbf{x}_i - \mathbf{c}_i)\|_2^2 + \frac{p - p^2}{n}\|\mathbf{x} - \mathbf{c}\|_2^2 - B\rho^2\epsilon_2 : \forall \mathbf{c} \in \mathcal{C}_q(\mathbf{u})\}. \tag{14}$$

respectively. Then, conditioned on $\mathcal{E}_1 \cap \mathcal{E}_2$ and noting that i) $\|\sum_{i=1}^{B}(\mathbf{x}_i - \tilde{\mathbf{x}}_i)\|_2^2 \leq B\|\mathbf{x} - \tilde{\mathbf{x}}\|_2^2$ and ii) $\frac{1}{\sqrt{nB}}\|\mathbf{x} - \tilde{\mathbf{x}}\|_2 \leq \delta$ and iii) for any $a, b \geq 0$, $\sqrt{a + b} \leq \sqrt{a} + \sqrt{b}$, from (10)- (12), we have

$$\sqrt{\frac{p - p^2}{nB}}\|\mathbf{x} - \hat{\mathbf{x}}_q\|_2 \leq \sqrt{p + (B - 1)p^2}\delta + \rho(\sqrt{\epsilon_1} + \sqrt{\epsilon_2}) + \sqrt{B}L2^{-q}\sqrt{\frac{k}{n}}, \tag{15}$$

or

$$\frac{1}{\sqrt{nB}}\|\mathbf{x}-\hat{\mathbf{x}}_q\|_2 \leq \sqrt{\frac{1+(B-1)p}{1-p}}\delta + \frac{\rho}{\sqrt{p(1-p)}}(\sqrt{\epsilon_1}+\sqrt{\epsilon_2}) + \sqrt{B}L2^{-q}\sqrt{\frac{k}{p(1-p)n}}. \tag{16}$$

On the other hand, by the triangle inequality, $\|\mathbf{x}-\hat{\mathbf{x}}\|_2 \leq \|\mathbf{x}-\hat{\mathbf{x}}_q\|_2 + \|\hat{\mathbf{x}}_q-\hat{\mathbf{x}}\|_2$. Therefore, combining (11) and (16), it follows that

$$\frac{1}{\sqrt{nB}}\|\mathbf{x}-\hat{\mathbf{x}}\|_2 \leq \sqrt{\frac{1+(B-1)p}{1-p}}\delta + \frac{\rho}{\sqrt{p(1-p)}}(\sqrt{\epsilon_1}+\sqrt{\epsilon_2})$$
$$+ \sqrt{B}L2^{-q}\sqrt{\frac{k}{p(1-p)n}} + L2^{-q}\sqrt{\frac{k}{nB}}. \tag{17}$$

To finish the proof, we need to set the parameters $(q,\epsilon_1,\epsilon_2)$ and bound $P(\mathcal{E}_1^c \cup \mathcal{E}_2^c)$. For a fixed $\mathbf{u} \in \mathbb{R}^{nB}$, $\|\mathbf{Hu}\|_2^2 = \sum_{j=1}^n U_j$, where $U_j = (\sum_{i=1}^B D_{ij}u_{ij})^2$. Note that

$$\mathrm{E}[U_j] = \mathrm{E}[\sum_{i=1}^B \sum_{i'=1}^B D_{ij}D_{i'j}u_{ij}u_{i'j}] = p^2(\sum_{i=1}^B u_{i,j})^2 + (p-p^2)\sum_{i=1}^B u_{i,j}^2. \tag{18}$$

Moreover, by the Cauchy-Schwartz inequality, $U_j \leq (\sum_{i=1}^B D_{ij}^2)\sum_{i=1}^B u_{ij}^2 \leq B\sum_{i=1}^B u_{ij}^2 \leq B^2(\|\mathbf{u}\|_\infty)^2$.

Therefore, since by assumption for all $\mathbf{x} \in \mathcal{Q}$, $\|\mathbf{x}\|_\infty \leq \frac{\rho}{2}$, using the Hoeffding's inequality, we have

$$P(\mathcal{E}_1^c) \leq \exp(-\frac{2n\epsilon_1^2}{B^2}). \tag{19}$$

Similarly, combining the Hoeffding's inequality and the union bound, we have

$$P(\mathcal{E}_2^c) \leq 2^{qk}\exp(-\frac{2n\epsilon_2^2}{B^2}). \tag{20}$$

Finally, given free parameters $\eta \in (0,1)$, setting the parameters as

$$\epsilon_1 = B\sqrt{\frac{\eta qk \ln 2}{2n}}, \epsilon_2 = B\sqrt{\frac{qk(1+\eta)\ln 2}{2n}}, \text{and } q = \lceil \log\log n \rceil,$$

we have

$$P((\mathcal{E}_1 \cap \mathcal{E}_2)^c) \leq 2^{-\eta qk+1} \leq 2^{-\eta k \log\log n+1}.$$

For the selected parameters, setting $\eta = 0.5$ and using $\frac{\ln 2}{2} < 1$ and $(\eta^{\frac{1}{4}}+(1+\eta)^{\frac{1}{4}}) \leq 2$, from (17), it follows that

$$\frac{1}{\sqrt{nB}}\|\mathbf{x}-\hat{\mathbf{x}}\|_2 \leq \sqrt{1+\frac{Bp}{1-p}}\delta + \frac{2\rho}{\sqrt{p(1-p)}}\left(\frac{kB^2\log\log n}{n}\right)^{\frac{1}{4}} + \frac{L}{\log n}\sqrt{\frac{k}{nB}}\left(\frac{B}{\sqrt{p(1-p)}}+1\right). \tag{21}$$

### A.3 Proof of Corollary 3.3

Let $f(p)$ denote the upper bound in Theorem 3.1. That is,

$$f(p) = \sqrt{1+\frac{Bp}{1-p}}\delta + \frac{1}{\sqrt{p(1-p)}}\upsilon_1 + \upsilon_2,$$

where $\epsilon_1$ and $\epsilon_2$ are defined as

$$\upsilon_1 = 2\rho\left(\frac{kB^2\log\log n}{n}\right)^{\frac{1}{4}} + \frac{L}{\log n}\sqrt{\frac{kB}{n}}, \quad \upsilon_2 = \frac{L}{\log n}\sqrt{\frac{k}{nB}},$$

and do not depend on $p$. On the other hand, $f(0) = f(1) = \infty$. Let $p^*$ denote the value of $p \in (0,1)$ that minimizes $f(p)$, note that

$$f'(p) = (1+\frac{Bp}{1-p})^{-1.5}\frac{B\delta}{(1-p)^2} - \frac{1}{2}(1-2p)(p-p^2)^{-1.5}\upsilon_1. \tag{22}$$

Note that on one hand $\lim_{p\to 0} f'(p) = -\infty$ and on the other hand $f'(\frac{1}{2}) > 0$, which implies that $p^*$ where $f'(p^*) = 0$ belongs to $(0,\frac{1}{2})$.

## A.4 Proof of Theorem 3.4

Let $\tilde{\mathbf{x}} = \arg\min_{\mathbf{c}=g_\theta(\mathbf{u}):\ \theta\in\mathbb{R}^k} \|\mathbf{x} - \mathbf{c}\|_2$ and

$$\hat{\mathbf{x}} = g_{\hat{\theta}}(\mathbf{u}) = \arg\min_{\mathbf{c}=g_\theta(\mathbf{u}):\ \theta\in\mathbb{R}^k} \|yv - \mathbf{H}\mathbf{c}\|_2^2, \quad \hat{\mathbf{x}}_q = g_{[\hat{\theta}]_q}(\mathbf{u}).$$

That is, $\hat{\mathbf{x}}_q$ denotes the reconstruction corresponding to the $q$-bit quantized version of $\hat{\theta}$. Following the same argument as the one used in the proof of Theorem 3.1, since $\mathbf{y} = \mathbf{H}\mathbf{x} + \mathbf{z}$, it follows that $\frac{1}{\sqrt{n}}\|\mathbf{x}-\tilde{\mathbf{x}}\|_2 \leq \delta$ and $\|\mathbf{H}(\mathbf{x}-\hat{\mathbf{x}})+\mathbf{z}\|_2 \leq \|\mathbf{H}(\mathbf{x}-\tilde{\mathbf{x}})+\mathbf{z}\|_2$. On the other hand, $\|\mathbf{H}(\mathbf{x}-\hat{\mathbf{x}})+\mathbf{z}\|_2^2 = \|\mathbf{H}(\mathbf{x}-\hat{\mathbf{x}})\|^2 + 2\langle\mathbf{z}, \mathbf{H}(\mathbf{x}-\hat{\mathbf{x}})\rangle + \|\mathbf{z}\|^2$, and $\|\mathbf{H}(\mathbf{x}-\tilde{\mathbf{x}})+\mathbf{z}\|_2^2 = \|\mathbf{H}(\mathbf{x}-\tilde{\mathbf{x}})\|^2 + 2\langle\mathbf{z}, \mathbf{H}(\mathbf{x}-\tilde{\mathbf{x}})\rangle + \|\mathbf{z}\|^2$. Therefore,

$$\|\mathbf{H}(\mathbf{x} - \hat{\mathbf{x}})\|^2 \leq \|\mathbf{H}(\mathbf{x} - \tilde{\mathbf{x}})\|^2 + 2|\langle\mathbf{z}, \mathbf{H}(\mathbf{x} - \hat{\mathbf{x}})\rangle| + 2|\langle\mathbf{z}, \mathbf{H}(\mathbf{x} - \tilde{\mathbf{x}})\rangle|. \tag{23}$$

Moreover, using the triangle inequality,

$$|\langle\mathbf{z}, \mathbf{H}(\mathbf{x} - \hat{\mathbf{x}})\rangle| \leq |\langle\mathbf{z}, \mathbf{H}(\mathbf{x} - \hat{\mathbf{x}}_q)\rangle| + |\langle\mathbf{z}, \mathbf{H}(\hat{\mathbf{x}}_q - \hat{\mathbf{x}})\rangle| \overset{(a)}{\leq} |\langle\mathbf{z}, \mathbf{H}(\mathbf{x} - \hat{\mathbf{x}}_q)\rangle| + \|\mathbf{z}\|_2\|\mathbf{H}(\hat{\mathbf{x}}_q - \hat{\mathbf{x}})\|_2,$$

where $(a)$ follows from Cauchy-Schwartz inequality. Therefore,

$$\|\mathbf{H}(\mathbf{x} - \hat{\mathbf{x}})\|^2 \leq \|\mathbf{H}(\mathbf{x} - \tilde{\mathbf{x}})\|^2 + 2|\langle\mathbf{z}, \mathbf{H}(\mathbf{x} - \hat{\mathbf{x}}_q)\rangle| + 2|\langle\mathbf{z}, \mathbf{H}(\mathbf{x} - \tilde{\mathbf{x}})\rangle| + \|\mathbf{z}\|_2\|\mathbf{H}(\hat{\mathbf{x}}_q - \hat{\mathbf{x}})\|_2. \tag{24}$$

Note that, by the triangle inequality, $\|\mathbf{H}(\mathbf{x} - \hat{\mathbf{x}})\|_2 \geq \|\mathbf{H}(\mathbf{x} - \hat{\mathbf{x}}_q)\|_2 - \|\mathbf{H}(\hat{\mathbf{x}}_q - \hat{\mathbf{x}})\|_2$, which implies that $\|\mathbf{H}(\mathbf{x} - \hat{\mathbf{x}})\|_2^2 \geq \|\mathbf{H}(\mathbf{x} - \hat{\mathbf{x}}_q)\|_2^2 - 2\mathbf{H}(\mathbf{x} - \hat{\mathbf{x}}_q)\|_2\|\mathbf{H}(\hat{\mathbf{x}}_q - \hat{\mathbf{x}})\|_2 + \|\mathbf{H}(\hat{\mathbf{x}}_q - \hat{\mathbf{x}})\|_2^2 \geq \|\mathbf{H}(\mathbf{x} - \hat{\mathbf{x}}_q)\|_2^2 - 2\|\mathbf{H}(\mathbf{x} - \hat{\mathbf{x}}_q)\|_2\|\mathbf{H}(\hat{\mathbf{x}}_q - \hat{\mathbf{x}})\|_2$. Therefore, combining this inequality with (24), it follows that

$$\begin{aligned}\|\mathbf{H}(\mathbf{x} - \hat{\mathbf{x}}_q)\|^2 \leq & \|\mathbf{H}(\mathbf{x} - \tilde{\mathbf{x}})\|^2 + 2|\langle\mathbf{z}, \mathbf{H}(\mathbf{x} - \hat{\mathbf{x}}_q)\rangle| + 2|\langle\mathbf{z}, \mathbf{H}(\mathbf{x} - \tilde{\mathbf{x}})\rangle| \\ & + (\|\mathbf{z}\|_2 + 2\|\mathbf{H}(\mathbf{x} - \hat{\mathbf{x}}_q)\|_2)\|\mathbf{H}(\hat{\mathbf{x}}_q - \hat{\mathbf{x}})\|_2.\end{aligned} \tag{25}$$

Similar to the proof of Theorem 3.1, for a fixed random initialization $\mathbf{u} \in \mathbb{R}^p$, define $\mathcal{C}_q(\mathbf{u}) = \{g_{[\theta]_q}(\mathbf{u}) : \theta \in [0, 1]^k\}$. Also, given $\epsilon_1, \epsilon_2, \epsilon_3 > 0$, and $\mathbf{x}, \tilde{\mathbf{x}} \in \mathbb{R}^{nB}$, define events $\mathcal{E}_1$ and $\mathcal{E}_2$ as (13) and (14), respectively. Moreover, define event $\mathcal{E}_3$ as

$$\mathcal{E}_3 = \{\frac{1}{n}\|\mathbf{H}(\mathbf{x} - \mathbf{c})\|_2^2 \leq \frac{p^2}{n}\|\sum_{i=1}^B (\mathbf{x}_i - \mathbf{c}_i)\|_2^2 + \frac{p - p^2}{n}\|\mathbf{x} - \mathbf{c}\|_2^2 + B\rho^2\epsilon_3 : \ \forall \mathbf{c} \in \mathcal{C}_q(\mathbf{u})\}, \tag{26}$$

Compared to the proof of Theorem 3.1, (24) involves three terms that involve Gaussian noise $\mathbf{z}$. For $\mathbf{c} \in \mathcal{C}_q(\mathbf{u})$, define $\phi(\mathbf{c})$ as

$$\phi(\mathbf{c}) \triangleq \langle\mathbf{z}, \mathbf{H}(\mathbf{x} - \mathbf{c})\rangle.$$

Conditioned on the mask $\mathbf{D}$, $\phi(\mathbf{c})$ is a zero-mean Gaussian random variable with

$$E[(\phi(\mathbf{c}))^2] = \sigma_z^2 \sum_{j=1}^n \left[\left(\sum_{i=1}^B D_{ij}(x_{ij} - c_{ij})\right)^2\right] = \sigma_z^2\|\sum \mathbf{D}_i(\mathbf{x}_i - \mathbf{c}_i)\|_2^2.$$

Let $t(\mathbf{c}) \triangleq \sigma_z\|\sum \mathbf{D}_i(\mathbf{x}_i - \mathbf{c}_i)\|_2$. Then, for any given $\epsilon_z > 0$,

$$\begin{aligned}P\left(|\phi(\mathbf{c})| \geq \sqrt{2n}\epsilon_z t(\mathbf{c})\right) &= \sum_{\mathbf{d}} P\left(\phi(\mathbf{c}) \geq \sqrt{2n}\epsilon_z t(\mathbf{c}) \mid \mathbf{D} = \mathbf{d}\right) P(\mathbf{D} = \mathbf{d}) \\ &\overset{(a)}{\leq} 2\sum_{\mathbf{d}} P(\mathbf{D} = \mathbf{d}) \exp(-\frac{2n\epsilon_z^2 t^2(\mathbf{c})}{2t^2(\mathbf{c})}) \\ &\leq 2\sum_{\mathbf{d}} P(\mathbf{D} = \mathbf{d}) \exp(-n\epsilon_z^2) \\ &\leq 2\exp(-n\epsilon_z^2),\end{aligned} \tag{27}$$

where $(a)$ follows because for any Gaussian random variable $G \sim \mathcal{N}(0, \sigma^2)$, applying the Chernoff bound, we have $\mathrm{P}(|G| > t) \leq 2\mathrm{e}^{-t^2/2\sigma^2}$. Given $\epsilon_{z1}, \epsilon_{z2}, \epsilon_{z3} > 0$, define events $\mathcal{E}_{z1}, \mathcal{E}_{z2}$ and $\mathcal{E}_{z3}$ as

$$\mathcal{E}_{z1} = \left\{ \left| \langle \mathbf{z}, \mathbf{H}(\mathbf{x} - \tilde{\mathbf{x}}) \rangle \right|_2 \leq \epsilon_{z1} \sqrt{2nB} \sigma_z \| \mathbf{H}(\mathbf{x} - \tilde{\mathbf{x}}) \|_2 \right\}, \tag{28}$$

$$\mathcal{E}_{z2} = \left\{ \left| \langle \mathbf{z}, \mathbf{H}(\mathbf{x} - \mathbf{c}) \rangle \right| \leq \epsilon_{z2} \sqrt{2nB} \sigma_z \| \mathbf{H}(\mathbf{x} - \mathbf{c}) \|_2 : \forall \mathbf{c} \in \mathcal{C}_q(\mathbf{u}) \right\}, \tag{29}$$

and

$$\mathcal{E}_{z3} = \left\{ \| \mathbf{z} \|_2^2 \leq n\sigma_z^2(1 + \epsilon_{z3}) \right\}, \tag{30}$$

respectively. From (27), noting that $|\mathcal{C}_q(\mathbf{u})| \leq 2^{qk}$, it follows that

$$\mathrm{P}\left(\mathcal{E}_{z1}^c\right) \leq 2\exp(-nB\epsilon_{z1}^2), \tag{31}$$

and

$$\mathrm{P}\left(\mathcal{E}_{z2}^c\right) \leq 2^{qk+1}\exp(-nB\epsilon_{z2}^2). \tag{32}$$

Define event $\mathcal{E}$ as $\mathcal{E} = \mathcal{E}_1 \cap \mathcal{E}_2 \cap \mathcal{E}_3 \cap \mathcal{E}_{z1} \cap \mathcal{E}_{z2}$ and

$$h(p) \triangleq p + (B-1)p^2.$$

Conditioned on $\mathcal{E}$, since by assumption $\frac{1}{\sqrt{nB}} \| \mathbf{x} - \tilde{\mathbf{x}} \|_2 \leq \delta$, we have

$$\left| \langle \mathbf{z}, \sum_{i=1}^B \mathbf{D}_i(\mathbf{x}_i - \tilde{\mathbf{x}}_i) \rangle \right| \leq \epsilon_{z1} \sqrt{2n\sigma_z^2((p + (B-1)p^2)nB\delta^2 + nB\rho^2\epsilon_1)},$$

or

$$\frac{1}{nB} \left| \langle \mathbf{z}, \sum_{i=1}^B \mathbf{D}_i(\mathbf{x}_i - \tilde{\mathbf{x}}_i) \rangle \right| \leq \epsilon_{z1}\sigma_z\sqrt{2h(p)}\delta + \sigma_z\rho\epsilon_{z1}\sqrt{2\epsilon_1}, \tag{33}$$

where the last line follows because for any $a, b > 0$, $\sqrt{a + b} \leq \sqrt{a} + \sqrt{b}$. Define

$$\Delta_q \triangleq \frac{1}{\sqrt{nB}} \| \mathbf{x} - \hat{\mathbf{x}}_q \|_2.$$

Then, similar to (33), conditioned on $\mathcal{E}$, since $\hat{\mathbf{x}}_q \in \mathcal{C}_q(\mathbf{u})$, it follows that

$$\frac{1}{nB} \left| \langle \mathbf{z}, \mathbf{H}(\mathbf{x} - \hat{\mathbf{x}}_q) \rangle \right| \leq \frac{\epsilon_{z2}}{nB} \sqrt{2nB\sigma_z^2((p + (B-1)p^2)\| \mathbf{x} - \hat{\mathbf{x}}_q \|_2^2 + nB\rho^2\epsilon_3)}$$

$$\leq \epsilon_{z2}\sigma_z\sqrt{2h(p)}\Delta_q + \sigma_z\rho\epsilon_{z2}\sqrt{2\epsilon_3}, \tag{34}$$

Conditioned on $\mathcal{E}$, combining (25), (33), and (34) it follows that

$$(p - p^2)\Delta_q^2 - \rho^2\epsilon_2 \leq h(p)\delta^2 + \rho^2\epsilon_1 + 2\sqrt{2}\epsilon_{z2}\sigma_z(\sqrt{h(p)}\Delta_q + \rho\sqrt{\epsilon_3})$$
$$+ 2\sqrt{2}\epsilon_{z1}\sigma_z(\sqrt{h(p)}\delta + \rho\sqrt{\epsilon_1})$$
$$+ (\sigma_z\sqrt{1 + \epsilon_{z3}} + 2\sqrt{B}(\sqrt{h(p)}\Delta_q + \rho\sqrt{\epsilon_3}))L2^{-q}\sqrt{\frac{k}{n}}, \tag{35}$$

where we have used $\| \mathbf{H}(\hat{\mathbf{x}}_q - \mathbf{x}) \|_2 \leq BL2^{-q}\sqrt{k}$ derived in (12). Note that since $\Delta_q \leq \rho$ and $h(p) \leq B$, the last term in (35) that corresponds to the quantization error can be bounded as

$$(\sigma_z\sqrt{1 + \epsilon_{z3}} + 2\sqrt{B}(\sqrt{h(p)}\Delta_q + \rho\sqrt{\epsilon_3}))\frac{BL2^{-q}k}{\sqrt{n}} \leq c_n,$$

where

$$c_n \triangleq (\sigma_z\sqrt{1 + \epsilon_{z3}} + 2\sqrt{B}\rho(\sqrt{B} + \sqrt{\epsilon_3}))L2^{-q}\sqrt{\frac{k}{n}},$$

and does not depend on $\Delta_q$. Rearranging the terms in (11) and letting

$$\epsilon_o \triangleq \rho^2(\epsilon_1 + \epsilon_2) + 2\sqrt{2}\rho\sigma_z(\epsilon_{z1}\sqrt{\epsilon_1} + \epsilon_{z2}\sqrt{\epsilon_3}).$$

it follows that

$$(p - p^2)\Delta_q^2 - \sigma_z\epsilon_{z2}\sqrt{8h(p)}\Delta_q \le h(p)\delta^2 + \epsilon_{z1}\sqrt{8h(p)}\delta\sigma_z + \epsilon_o + c_n. \tag{36}$$

Therefore,

$$(p - p^2)\left(\Delta_q - \frac{\sigma_z\epsilon_{z2}\sqrt{2h(p)}}{p(1-p)}\right)^2 \le \frac{\sigma_z^2\epsilon_{z2}^2 h(p)}{2p(1-p)} + h(p)\delta^2 + \epsilon_{z1}\sqrt{8h(p)}\delta\sigma_z + \epsilon_o + c_n. \tag{37}$$

This implies that

$$\Delta_q \le \frac{\sigma_z\epsilon_{z2}\sqrt{2h(p)}}{p(1-p)} + \sqrt{\frac{1}{p(1-p)}\left(\frac{\sigma_z^2\epsilon_{z2}^2 h(p)}{2p(1-p)} + h(p)\delta^2 + \epsilon_{z1}\sqrt{8h(p)}\delta\sigma_z + \epsilon_o + c_n\right)}. \tag{38}$$

Therefore, since i) $\|\mathbf{x} - \hat{\mathbf{x}}\|_2 \le \|\mathbf{x} - \hat{\mathbf{x}}_q\|_2 + \|\hat{\mathbf{x}}_q - \hat{\mathbf{x}}\|_2$ and ii) from (11), $\|\hat{\mathbf{x}}_q - \hat{\mathbf{x}}\|_2 \le L2^{-q}\sqrt{k}$, we have

$$\frac{1}{\sqrt{nB}}\|\mathbf{x} - \hat{\mathbf{x}}\|_2 \le \frac{\sigma_z\epsilon_{z2}\sqrt{2h(p)}}{p(1-p)} + \sqrt{\frac{1}{p(1-p)}\left(\frac{\sigma_z^2\epsilon_{z2}^2 h(p)}{2p(1-p)} + h(p)\delta^2 + \epsilon_{z1}\sqrt{8h(p)}\delta\sigma_z + \epsilon_o + c_n\right)}$$

$$+ L2^{-q}\sqrt{\frac{k}{nB}}. \tag{39}$$

Using $\sqrt{1 + \alpha} \le 1 + 2\alpha$, for $\alpha > 0$, and noting that $\frac{h(p)}{p(1-p)} = 1 + Bp/(1-p)$, we have

$$\frac{1}{\sqrt{nB}}\|\mathbf{x} - \hat{\mathbf{x}}\|_2 \le \frac{\sigma_z\epsilon_{z2}\sqrt{2h(p)}}{p(1-p)} + \delta\sqrt{1 + \frac{Bp}{1-p}}\left(1 + \frac{\sigma_z^2\epsilon_{z2}^2}{p(1-p)\delta^2} + 4\epsilon_{z1}\sigma_z\sqrt{\frac{2}{h(p)\delta^2}} + \frac{2(\epsilon_o + c_n)}{\delta^2 h(p)}\right)$$

$$+ L2^{-q}\sqrt{\frac{k}{nB}}. \tag{40}$$

Next we set the parameters by analyzing the probability event $\mathcal{E}$. Applying the union bound,

$$\mathrm{P}(\mathcal{E}^c) = \mathrm{P}((\mathcal{E}_1 \cap \mathcal{E}_2 \cap \mathcal{E}_3 \cap \mathcal{E}_{z1} \cap \mathcal{E}_{z2} \cap \mathcal{E}_{z3})^c)$$

$$\le \exp(-\frac{2n\epsilon_1^2}{B^2}) + 2^{qk}\left(\exp(-\frac{2n\epsilon_2^2}{B^2}) + \exp(-\frac{2n\epsilon_3^2}{B^2})\right) + e^{-\frac{n}{2}(\epsilon_{z3} - \log(1 + \epsilon_{z3}))}$$

$$+ 2\exp(-nB\epsilon_{z1}^2) + 2^{qk+1}\exp(-nB\epsilon_{z2}^2)), \tag{41}$$

where we have used Lemma A.1 to bound $\mathrm{P}(\mathcal{E}_{z3}^c)$. Similar to the proof of Theorem 3.1, given free parameters $\eta_1, \eta_2 \in (0, 1)$, let

$$\epsilon_1 = B\sqrt{\frac{\eta_1 qk \ln 2}{2n}}, \epsilon_2 = \epsilon_3 = B\sqrt{\frac{qk(1 + \eta_1) \ln 2}{2n}}, \text{and } q = \lceil \log \log n \rceil.$$

Also, set

$$\epsilon_{z1} = \sqrt{\frac{\eta_2 qk \ln 2}{nB}}, \ \epsilon_{z2} = \sqrt{\frac{(1 + \eta_2)qk \ln 2}{nB}}, \text{ and } \epsilon_{z3} = 1.$$

Following a similar argument as the one used in the proof of Theorem 3.1, it follows that

$$P(\mathcal{E}^c) \le 2^{-\eta_1 qk+2} + 2^{-\eta_2 qk+1} + e^{-0.3n} \le 2^{-\eta_1 k \log \log n+2} + 2^{-\eta_2 k \log \log n+1} + e^{-0.3n}.$$

Setting $\eta_1 = \eta_2 = 0.5$, it is straight forward to see that

$$\epsilon_o \le \rho^2\sqrt{\frac{qkB^2}{n}} + \rho\sigma_z(\frac{q^3 k^3}{B^2 n^3})^{\frac{1}{4}} \stackrel{(a)}{\le} \rho^2\frac{1}{(\log n)^{0.25}} + \rho\sigma_z(\frac{k^3}{B^2(n \log n)^3})^{\frac{1}{4}}, \tag{42}$$

where (a) follows from (5). Also, for the selected parameters and noting that $B \ge 1$,

$$c_n \le (\sigma_z\sqrt{2} + 2B\rho(1 + (\frac{qk}{n})^{0.25})\frac{L}{\log n}\sqrt{\frac{k}{n}} \le (\sigma_z\sqrt{2} + 4B\rho)\frac{L}{\log n}, \tag{43}$$

Therefore, $c_n = O(\frac{1}{\log n})$. Moreover,

$$\epsilon_{z1} \leq \sqrt{\frac{k \log \log n}{nB}}, \ \epsilon_{z2} \leq \sqrt{\frac{2k \log \log n}{nB}}.$$

Therefore, from (39), and using i) $\sqrt{\sum_i a_i} \leq \sum_i \sqrt{a_i}$, for $a_i \geq 0$, ii) $h(p) \leq B$ for all $p \in [0, 1]$ and iii) $B \geq 1$ (which implies from our assumption that $k(\log n)(\log \log n) < n$), it follows that

$$\frac{1}{\sqrt{nB}}\|\mathbf{x} - \hat{\mathbf{x}}\|_2 \leq \delta \sqrt{\frac{1 + (B-1)p}{1-p}} + \frac{3\sigma_z}{p(1-p)}\sqrt{\frac{1}{\log n}} + (\frac{8}{\log n})^{\frac{1}{4}}\sqrt{\frac{\delta\sigma_z}{p(1-p)}}$$

$$+ \sqrt{\frac{1}{p(1-p)}}\Big(\rho\frac{1}{(\log n)^{\frac{1}{8}}} + \sqrt{\rho\sigma_z}(\frac{1}{B(\log n)^3})^{\frac{1}{4}} + \sqrt{c_n}\Big) + o(\frac{1}{\log n}),$$

or, rearranging the term in (1), we have

$$\frac{1}{\sqrt{nB}}\|\mathbf{x} - \hat{\mathbf{x}}\|_2 \leq \delta\sqrt{1 + \frac{Bp}{1-p}} + \frac{3\sigma_z}{p(1-p)}\sqrt{\frac{1}{\log n}}$$

$$+ (\frac{8}{\log n})^{\frac{1}{4}}\sqrt{\frac{\delta\sigma_z}{p(1-p)}}\Big(1 + \frac{1}{B8^{1/4}\sqrt{\log n}}\Big)$$

$$+ \sqrt{\frac{1}{p(1-p)}}(\rho\frac{1}{(\log n)^{\frac{1}{8}}} + \sqrt{c_n}) + o(\frac{1}{\log n}), \tag{44}$$

which yields the desired result.

## A.5 Proof of Theorem 3.5

Following the steps of the proof of Theorem 3.4, we can derive (25), i.e., $\|\mathbf{H}(\mathbf{x} - \hat{\mathbf{x}}_q)\|^2 \leq \|\mathbf{H}(\mathbf{x} - \tilde{\mathbf{x}})\|^2 + 2|\langle \mathbf{z}, \mathbf{H}(\mathbf{x} - \hat{\mathbf{x}}_q)\rangle| + 2|\langle \mathbf{z}, \mathbf{H}(\mathbf{x} - \tilde{\mathbf{x}})\rangle| + (\|\mathbf{z}\|_2 + 2\|\mathbf{H}(\mathbf{x} - \hat{\mathbf{x}}_q)\|_2)\|\mathbf{H}(\hat{\mathbf{x}}_q - \hat{\mathbf{x}})\|_2$. We defined events $\mathcal{E}_1, \mathcal{E}_2, \mathcal{E}_3, \mathcal{E}_{z1}, \mathcal{E}_{z2}$ and $\mathcal{E}_{z3}$ as before. Note that conditioned on $\mathcal{E}_2$,

$$\frac{1}{n}\|\mathbf{H}(\mathbf{x} - \hat{\mathbf{x}}_q)\|_2^2 \geq \frac{p^2}{n}\|\sum_{i=1}^{B}(\mathbf{x}_i - \hat{\mathbf{x}}_i)\|_2^2 + \frac{p - p^2}{n}\|\mathbf{x} - \hat{\mathbf{x}}\|_2^2 - B\rho^2\epsilon_2.$$

In the proof of Theorems 3.1 and 3.4, to apply this inequality and bound the error, we ignored the positive term $\frac{p^2}{n}\|\sum_{i=1}^{B}(\mathbf{x}_i - \hat{\mathbf{x}}_i)\|_2^2$ and used $\frac{1}{n}\|\mathbf{H}(\mathbf{x} - \hat{\mathbf{x}}_q)\|_2^2 \geq \frac{p-p^2}{n}\|\mathbf{x} - \hat{\mathbf{x}}\|_2^2 - B\rho^2\epsilon_2$. Here, instead we ignore $\frac{p-p^2}{n}\|\mathbf{x} - \hat{\mathbf{x}}\|_2^2$ and conditioned on $\mathcal{E}_2$, we use

$$\frac{1}{n}\|\mathbf{H}(\mathbf{x} - \hat{\mathbf{x}}_q)\|_2^2 \geq \frac{p^2}{n}\|\sum_{i=1}^{B}(\mathbf{x}_i - \hat{\mathbf{x}}_i)\|_2^2 - B\rho^2\epsilon_2. \tag{45}$$

Define $\mathcal{E} = \mathcal{E}_1 \cap \mathcal{E}_2 \cap \mathcal{E}_3 \cap \mathcal{E}_{z1} \cap \mathcal{E}_{z2}$ similar to the proof of Theorem 3.4. Following the steps used in deriving (36) in the proof of Theorem 3.4, and applying i) the lower bound in (45) and ii) $\Delta_q \leq \rho$, it follows that, conditioned on $\mathcal{E}$,

$$\frac{p^2}{nB}\|\sum_{i=1}^{B}(\mathbf{x}_i - \hat{\mathbf{x}}_{q,i})\|_2^2 \leq h(p)\delta^2 + (\epsilon_{z1}\delta + \rho\epsilon_{z2})\sigma_z\sqrt{8h(p)} + \epsilon_o + c_n. \tag{46}$$

Let

$$\bar{\mathbf{x}} = \frac{1}{B}\sum_{i=1}^{B}\mathbf{x}_i, \quad \bar{\hat{\mathbf{x}}}_q = \frac{1}{B}\sum_{i=1}^{B}\hat{\mathbf{x}}_{q,i}, \text{and} \quad \bar{\hat{\mathbf{x}}} = \frac{1}{B}\sum_{i=1}^{B}\hat{\mathbf{x}}_i,$$

and

$$\bar{\Delta}_q^2 = \frac{1}{n}\|\bar{\mathbf{x}} - \bar{\hat{\mathbf{x}}}_q\|_2^2, \quad \bar{\Delta}^2 = \frac{1}{n}\|\bar{\mathbf{x}} - \bar{\hat{\mathbf{x}}}\|_2^2.$$

Using these definitions, (46) can be written as

$$\bar{\Delta}_q^2 \leq \frac{1}{p^2 B}\Big(h(p)\delta^2 + (\epsilon_{z1}\delta + \rho\epsilon_{z2})\sigma_z\sqrt{8h(p)} + \epsilon_o + c_n\Big). \tag{47}$$

Noting that $\frac{1}{p^2 B}h(p) = \frac{B-1}{B} + \frac{1}{pB} \leq 1 + \frac{1}{pB}$,

$$\bar{\Delta}_q^2 \leq (1 + \frac{1}{pB})\delta^2 + \frac{1}{p^2 B}\Big((\epsilon_{z1}\delta + \rho\epsilon_{z2})\sigma_z\sqrt{8h(p)} + \epsilon_o + c_n\Big). \tag{48}$$

Using Cauchy-Schwarz inequality, $\|\sum_{i=1}^B(\hat{\mathbf{x}}_i - \hat{\mathbf{x}}_{q,i})\|_2^2 \leq B\|\hat{\mathbf{x}} - \hat{\mathbf{x}}_q\|_2^2$ and combining it with (11), it follows that

$$\|\sum_{i=1}^B(\hat{\mathbf{x}}_i - \hat{\mathbf{x}}_{q,i})\|_2^2 \leq B(L2^{-q}\sqrt{k})^2. \tag{49}$$

or

$$\frac{1}{\sqrt{n}}\|\hat{\bar{\mathbf{x}}} - \hat{\bar{\mathbf{x}}}_q\|_2 \leq L2^{-q}\sqrt{\frac{k}{nB}}. \tag{50}$$

Therefore, using the triangle inequality as $\|\bar{\mathbf{x}} - \hat{\bar{\mathbf{x}}}\|_2 \leq \|\bar{\mathbf{x}} - \hat{\bar{\mathbf{x}}}_q\|_2 + \|\hat{\bar{\mathbf{x}}}_q - \hat{\bar{\mathbf{x}}}\|_2$, it follows from (48) and (50) that

$$\frac{1}{\sqrt{n}}\|\bar{\mathbf{x}} - \hat{\bar{\mathbf{x}}}\|_2 \leq L2^{-q}\sqrt{\frac{k}{nB}} + \sqrt{(1 + \frac{1}{pB})\delta^2 + \frac{1}{p^2 B}\Big((\epsilon_{z1}\delta + \rho\epsilon_{z2})\sigma_z\sqrt{8h(p)} + \epsilon_o + c_n\Big)}$$

$$\leq L2^{-q}\sqrt{\frac{k}{nB}} + \delta\sqrt{1 + \frac{1}{pB}} + \frac{1}{p\sqrt{B}}\sqrt{\Big((\epsilon_{z1} + \epsilon_{z2})\rho\sigma_z\sqrt{8h(p)} + \epsilon_o + c_n\Big)} \tag{51}$$

where the last line follows from $\delta \leq \rho$, and $\sqrt{a+b} \leq \sqrt{a} + \sqrt{b}$, for $a, b \geq 0$. We set the parameters as in the proof of Theorem 3.5 as

$$\epsilon_1 = B\sqrt{\frac{0.5qk\ln 2}{2n}}, \quad \epsilon_2 = \epsilon_3 = B\sqrt{\frac{1.5qk\ln 2}{2n}}, \quad \text{and } q = \lceil \log\log n \rceil.$$

Also, set

$$\epsilon_{z1} = \sqrt{\frac{0.5qk\ln 2}{nB}}, \quad \epsilon_{z2} = \sqrt{\frac{1.5qk\ln 2}{nB}}, \text{ and } \epsilon_{z3} = 1.$$

Then, using the bounds in (42) and (43), it follows from (51)

$$\frac{1}{\sqrt{n}}\|\bar{\mathbf{x}} - \hat{\bar{\mathbf{x}}}\|_2 \leq \delta\sqrt{1 + \frac{1}{pB}} + \frac{1}{p}\sqrt{\frac{2\rho\sigma_z}{B}}\Big(\frac{k\log\log n}{n}h(p)\Big)^{\frac{1}{4}} + \frac{1}{p\sqrt{B}}\upsilon_n + \frac{L}{\log n}\sqrt{\frac{k}{nB}}, \tag{52}$$

where $\upsilon_n = O(\frac{1}{(\log n)^{\frac{1}{8}}})$ and does not depend on $p$.

# B  Setup of SCI-BDVP

## B.1  SCI-BDVP: Descent step

As explained in the paper, to solve the optimization described in (3), we employ the PGD algorithm, with an additional skip connection. The details of the projections step using bagged-DVP and also the skip connection are described in Section 4. Here, we review the descent step, as we employ two different operators depending on whether the measurements are noisy or noiseless.

**Descent step:**

- For noise-free measurements, we use GAP update rule [43]:

$$\mathbf{x}_{t+1}^G = \mathbf{x}_t + \mu\mathbf{H}^\top(\mathbf{H}\mathbf{H}^\top)^{-1}(\mathbf{y} - \mathbf{H}\mathbf{x}_t), \tag{53}$$

- For nosiy measurement, we use gradient descent (GD):

$$\mathbf{x}_{t+1}^G = \mathbf{x}_t + \mu \mathbf{H}^\top (\mathbf{y} - \mathbf{H}\mathbf{x}_t), \tag{54}$$

In both cases $\mu$ denotes the step size.

Compared to the GD, if $\mu = 1$, GAP, at each iteration, projects the current estimate $\mathbf{x}^{(t)}$ onto the $\mathbf{y} = \mathbf{H}\mathbf{x}$ hyperplane. Note that due to the special structure of the sensing matrix $\mathbf{H}$, $\mathbf{H}\mathbf{H}^\top$ is a diagonal matrix and therefore it is straightforward to compute its inverse, as required by GAP.

In our experiments, we found that in the case of noise-free measurements, the GAP update rule consistently showed better convergence compared to GD. Therefore, we adopted GAP update rule for the case of noise-free measurements. However, for noisy measurements, even the true signal does not lie on $\mathbf{y} = \mathbf{H}\mathbf{x}$ hyperplane, and therefore, application of GAP is no longer theoretically founded. Hence, for all experiments done for noisy measurements, we use the classic GD update rule.

In summary, Algorithm 1 below shows the steps of SCI-BDVP.

---

**Algorithm 1** SCI-BDVP

---

**Require:** measurement $\mathbf{y}$, mask $\mathbf{H}$
  1: Initial $\mathbf{x}_0 = \mathbf{H}^\top \mathbf{y}$.
  2: **for** $t = 1, \ldots, T$ **do**
  3:    **Descent step**
  4:      Update $\mathbf{x}_t^G$ with Eq. (53) or (54).
  5:    **Projection step**
  6:      Generate $\mathbf{x}_t^P$ as the output of bagged-DVP (refer to Fig. 2)
  7:      Update $\mathbf{x}_t$ with $\mathbf{x}_t = \alpha \mathbf{x}_t^G + (1 - \alpha)\mathbf{x}_t^P$
  8: **end for**
  9: **Output:** Reconstructed signal $\hat{\mathbf{x}} = \mathbf{x}_T$.

---

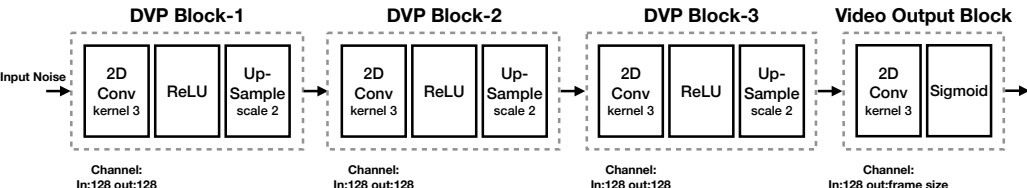

Figure 6: Network structure of DVP we use in SCI-BDVP.

## B.2 Implementation details

In the projection step, we use the same structure design for bagged-DVP, for both noiseless and noisy measurements. Inspired by deep decoder structure [6], we design the neural nets, using three DVP blocks and one video output block shown in Figure 6. Each DVP block is composed of *Upsample*, *ReLU* and *Conv* blocks. The output block only contains the *Conv* block. Here, we use *Conv* $3 \times 3$ and the number of channels are fixed to $128$. Lastly, the input $\mathbf{u}$ of each DVP (described in DVP function $g_\theta(\mathbf{u})$) is generated independently using a Uniform distribution, $U(0, 1)$.

Since the input video consists of $B$ $256 \times 256$ frames, we choose three DVP structures, one with $16$ $64 \times 64 \times B$ patches, one with $4$ $128 \times 128 \times B$ patches, and one with a single $256 \times 256 \times B$ frame. For each size of the patches, we perform mirror padding to augment the each patch with the size of $h/8$, since it is square patch, where $h$ represent the height of the padded patch. And for each patch of each estimate, we train the separate DVP module.

The hyperparameters are set as follows: the learning rate of the DVPs is set to 0.01; weight $\omega = 0.1$ for measurement loss term $\omega \|\mathbf{y}_i - \mathbf{H}_i g_\theta^{k,i}(\mathbf{u}_{k,i})\|_2^2$ in Figure 2. For noise free case, GAP, we set the step size $\mu = 1.0$, and $\mu = 0.1$ for the noisy simple GD case.

Table 4: Number of inner and outer iterations for training SCI-BDVP for different datasets and different estimates.

|  | Iterations | Kobe | Traffic | Runner | Drop | Crash | Aerial |
|---|---|---|---|---|---|---|---|
| No noise | Inner iteration-64 | 2000 | 700 | 2000 | 2000 | 700 | 700 |
|  | Inner iteration-128 | 2000 | 700 | 2000 | 2000 | 700 | 700 |
|  | Inner iteration-256 | 4000 | 1400 | 4000 | 4000 | 1400 | 1400 |
|  | Outer iteration | 75 | 35 | 75 | 75 | 35 | 35 |
| Noisy | Inner iteration-64 | 900 | 900 | 900 | 900 | 900 | 900 |
|  | Inner iteration-128 | 900 | 900 | 900 | 900 | 900 | 900 |
|  | Inner iteration-256 | 1800 | 1800 | 1800 | 1800 | 1800 | 1800 |
|  | Outer iteration | 35 | 35 | 35 | 35 | 35 | 35 |

Table 5: Time complexity over different methods on one 8-frame benchmark video block.

|  | Methods | Time (min.) |
|---|---|---|
| No noise | PnP-DIP [27] | 18 |
|  | Factorized-DVP [26] | 15 |
|  | Simple-DVP(E2E) | 10 |
|  | SCI-BDVP | 35 or 220 |
| No noise | PnP-DIP [27] | 18 |
|  | Factorized-DVP [26] | — |
|  | Simple-DVP(E2E) | 10 |
|  | SCI-BDVP | 40 |

## B.3 Time/computational complexity

Implementing SCI-BDVP involves outer loop iterations (described in Algorithm 1) and also inner loop iterations for training DIPs. Table 4 presents average number of inner loop iterations used for different patch sizes $(64, 128, 256)$ of various videos, and the number of outer loop iterations. Detailed time consumption for each patch level computation is recorded in Table 3. A comparison across different UNN-based methods is provided in Table 5. All comparisons are performed on a single NVIDIA RTX 4090. It is important to note that training a bagged DIP requires training multiple separate DIPs. This process can be **readily parallelized**, which is expected to significantly speed up the algorithm. We plan to explore this direction to optimize the algorithm's efficiency in future work. Lastly, making a direct comparison among all methods is challenging because, for supervised methods, the main time is spent in training, whereas, for unsupervised methods, the main time is spent on training the UNNs. This is an expected trade-off for requiring no training data and achieving a robust solution.

## C Additional studies

### C.1 Mask optimization

In this paper, we explored binary masks that are generated i.i.d. $\mathrm{Bern}(p)$. Figures 4 and 5 show the effect of probability $p$ on the performance of SCI-BDVP (GAP) (noiseless measurements) and SCI-BDVP (GD) (noisy measurements), respectively. In this section, we perform similar investigation of the effect of $p$ on the performance of other SCI methods, namely, PnP-FastDVD [18] and PnP-DIP [27]. Figure 7 shows the results corresponding to PnP-FastDVD. It can be observed that the trends are consistent with the performance of our proposed SCI-BDVP (GAP): i) for noiseless measurements, the optimal performance is achieved at $p^* < 0.5$, ii) for noisy measurements, $p^*$ is an increasing function of $\sigma_z$.

On the other hand, Figure 8 shows the performance achieved by PnP-DIP proposed in [27]. It can be observed that the reconstruction performance is not longer a smooth function of $p$, unlike the performance of SCI-BDVP and PnP-FastDVD. We believe that the reason lies in the the limitations of UNNs (or DIP) explained before, such as overfitting and instability. In the next section, we further explore this issue and explain how bagging can address the problem.

Also, we include the detailed PSNR and SSIM results using SCI-BDVP on different measurement noise level in Table 6. We can find when $\sigma = 50$, the effect of the mask optimization will improve the overall result around $1$ dB in PSNR, $0.1$ in SSIM. Here, all the algorithm settings are kept intact, and the only variation is in the mask non-zero probability $p$ varies between $0.5$ to $0.7$. This further highlights the stability of the proposed SCI-BDVP solution.

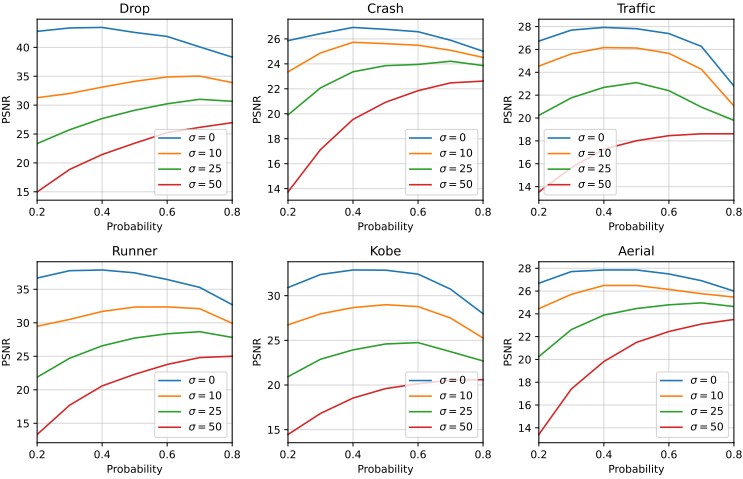

Figure 7: PSNR of $\|\mathbf{x} - \hat{\mathbf{x}}\|$ under different mask generated from $\mathrm{Bern}(p)$ of different **measurement noise** level using **baseline method** (PnP-FastDVD[18] with GAP gradient descent algorithm).

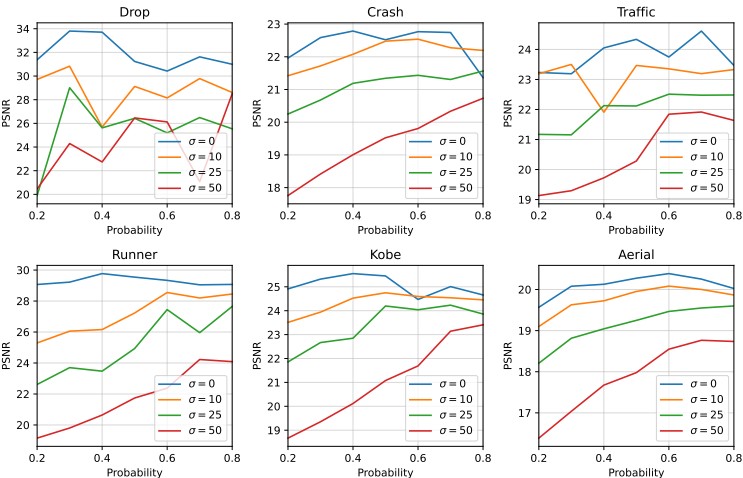

Figure 8: PSNR of $\|\mathbf{x} - \hat{\mathbf{x}}\|$ under different mask generated from $\mathrm{Bern}(p)$ of different **measurement noise** level using **baseline method** (PnP-DIP[27] with ADMM gradient descent algorithm).

## C.2  Effect of bagging

We discussed how bagging can help address the DIP (or DVP) overfitting issue and provide a robust projection module, which can robustly capture the source structure, without any training data. Figure 9 shows the impact of bagging on the performance of SCI-BDVP in Section 4, by comparing it with the performances of different SCI-DVP solutions, where instead of a bagged version, we used a simple DVP for projection.

Figure 10 shows the qualitative reconstruction results of our proposed SCI-BDVP in comparision with the non-bagged version, SCI-DVP. An expected, the results show that using bagging improves the reconstruction quality, in both noise-free and noisy cases.

Table 6: **Detailed mask optimization effect on reconstruction with SCI-BDVP.** PSNR (dB) (left entry) and SSIM (right entry) of the reconstruction results on different videos. (Reg.) represent reconstruction on using fixed regular binary mask, $D_{ij} \sim \text{Bern}(0.5)$. (OPT.) represents model tested on fixed optimized mask.

| Noise Level | Mask Choice | Kobe | Traffic | Runner | Drop | Crash | Aerial | Average |
|---|---|---|---|---|---|---|---|---|
| $\sigma = 0$ | **B-DVP (GAP) Reg.** | 28.42, 0.886 | 22.84, 0.779 | 34.32, 0.954 | 40.76, 0.986 | 24.96, 0.851 | 25.16, 0.837 | 29.41, 0.882 |
| | **B-DVP (GAP) Opt.** | 28.73, 0.891 | 23.47, 0.791 | 35.00, 0.958 | 41.33, 0.988 | 25.66, 0.860 | 25.52, 0.841 | 29.95, 0.888 |
| $\sigma = 10$ | **B-DVP (PGD) Reg.** | 26.39, 0.805 | 22.66, 0.740 | 31.15, 0.916 | 35.03, 0.962 | 25.57, 0.835 | 25.62, 0.817 | 27.73, 0.846 |
| | **B-DVP (PGD) Opt.** | 26.48, 0.812 | 22.72, 0.743 | 31.25, 0.914 | 35.30, 0.963 | 25.50, 0.831 | 25.47, 0.814 | 27.78, 0.846 |
| $\sigma = 25$ | **B-DVP (PGD) Reg.** | 25.89, 0.775 | 22.23, 0.718 | 30.31, 0.895 | 34.17, 0.954 | 25.33, 0.821 | 25.47, 0.796 | 27.23, 0.827 |
| | **B-DVP (PGD) Opt.** | 25.89, 0.775 | 22.23, 0.718 | 30.31, 0.895 | 34.17, 0.954 | 25.33, 0.821 | 25.47, 0.796 | 27.23, 0.827 |
| $\sigma = 50$ | **B-DVP (PGD) Reg.** | 23.34, 0.640 | 20.56, 0.611 | 25.11, 0.693 | 29.86, 0.889 | 23.43, 0.693 | 22.97, 0.638 | 24.21, 0.694 |
| | **B-DVP (PGD) Opt.** | 23.71, 0.685 | 21.02, 0.649 | 26.52, 0.812 | 31.85, 0.930 | 24.26, 0.772 | 24.10, 0.735 | 25.24, 0.764 |

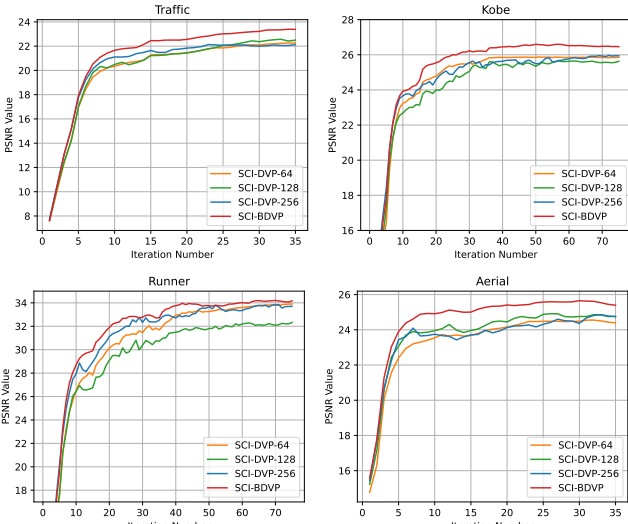

Figure 9: **Effect of bagging.** Reconstruction PSNR of SCI-BDVP vs. SCI-DVP, where in each SCI-DVP a separate DVP is employed (noise-free measurements).

Finally, to further highlight the impact of the bagging operation, here we explore the performance of the proposed bagged DVP solution for the classic inverse problem of denoising from additive Gaussian noise. Figure 12 shows the denoising performance of the proposed bagged-DVP solution (refer to Figure 2) in denoising $\mathbf{x}$ from measurements $\mathbf{y} = \mathbf{x} + \mathbf{z}$, where $\mathbf{z}$ is generated i.i.d. $\mathcal{N}(0, \sigma_z^2)$. We compare the performance of bagged-DVP with the three DVP structures that are used as the building components of our bagged-DVP. As explained earlier, each of these DVPs operates at a different patch size. It can be observed that BDVP consistently outperforms the individual DVPs and shows a much more smooth convergence behaviour. Given that BDVP only averages the outputs of the three individual DVPs, the observed gain suggests the independence of the estimates (at least partially), which leads to the observed gain.

## C.3 Effect of averaging coefficient

We explore the effect of coefficient $\alpha$ used in combining the outputs from gradient descent and projection steps at time $t$,

$$\mathbf{x}_t = \alpha \mathbf{x}_t^G + (1 - \alpha) \mathbf{x}_t^P,$$

In Figure 11 left panel, it can be observed that without the skip connection, the performance drops by around 4 dB. However, for $\alpha \in [0.1, 0.7]$ the performance is stable and does not vary considerably as $\alpha$ changes. On the other extreme case when $\alpha = 1$, when there is no projection, as expected, the performance severely degrades.

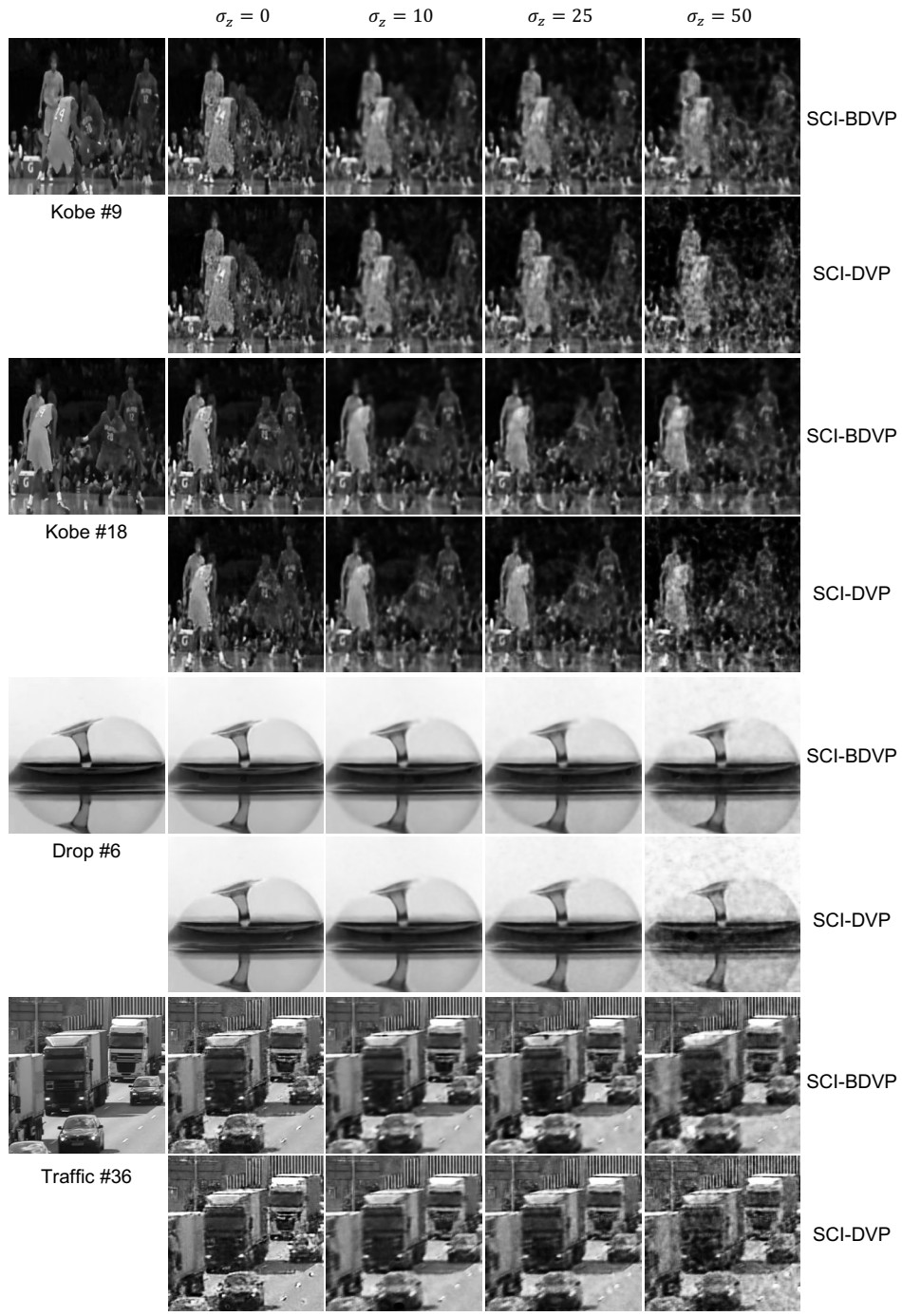

## C.4 Effect of reconstruction loss coefficient

We explore the effect of coefficient in reconstruction measurement loss, the second term of the loss function in Figure 2

$$\omega\|y - Ag_\theta(\mathbf{u})\|_2.$$

In the middle and right panel of Figure 11, we can find that for noise-free case the reconstruction results is not sensitive to the change of $\omega$. However, in the noisy case, when we increase the $\omega$ the reconstruction will drop and if not including the measurement loss term, $\omega = 0$, the results will also

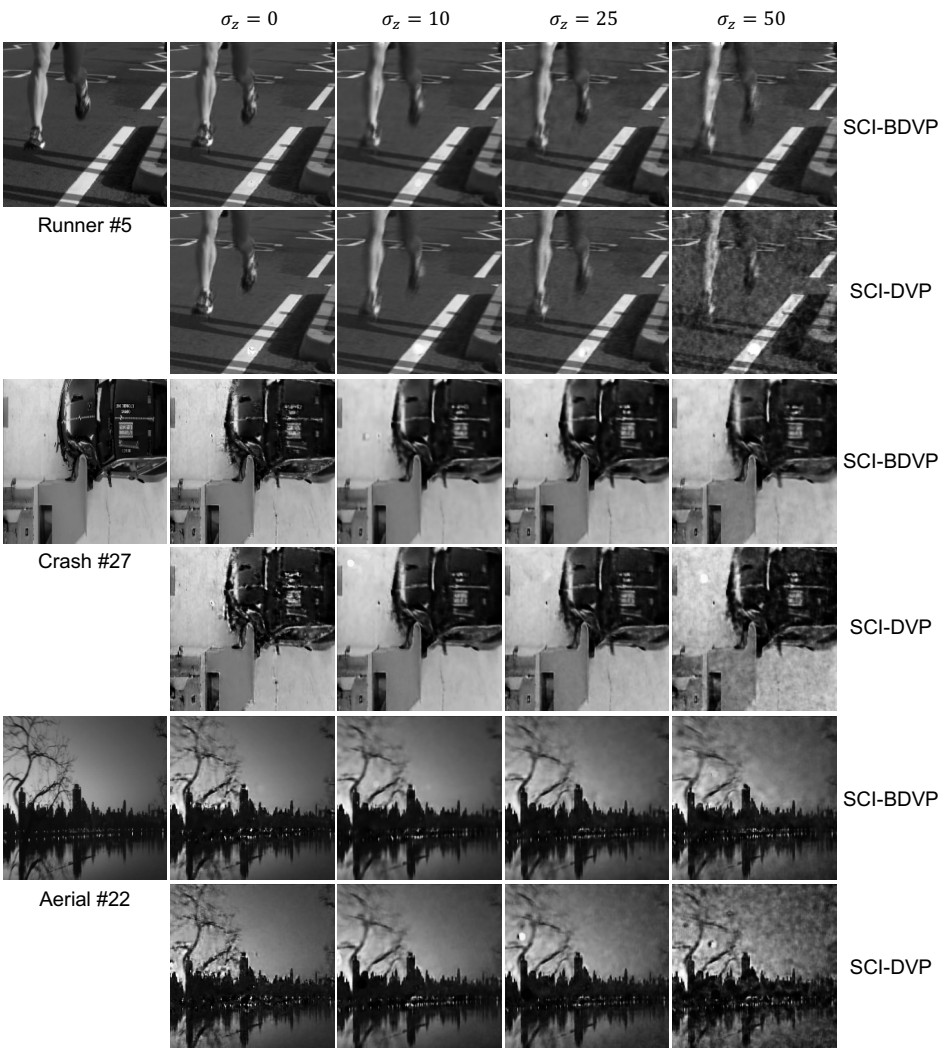

Figure 10: **Reconstruction results of SCI-BDVP vs. SCI-DVP.** (leftmost images are clean frames).

drop. This is due to in noisy case, the measurement **y** is no longer the actual measurement, but we still need some information from corrupted **y** to boost the reconstruction results.

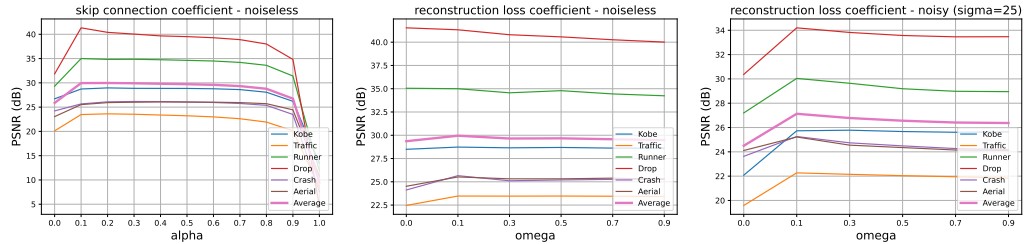

Figure 11: (Left) Effect of skip connection coefficient $\alpha$ (noiseless measurements). (Middle) Effect of reconstruction loss coefficient $\omega$ (noiseless measurements). (Right) Effect of reconstruction loss coefficient $\omega$ effect (noisy measurements) ($\sigma = 25$).

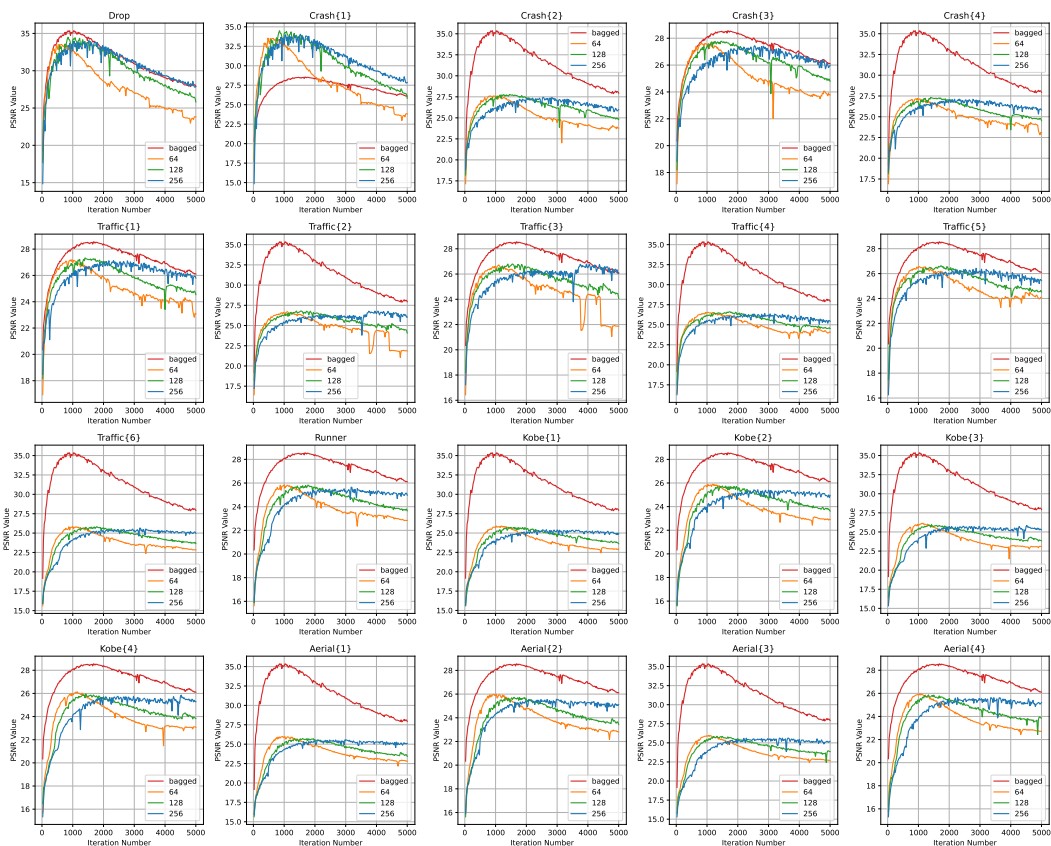

Figure 12: **Unsupervised video denoising – Effect of bagging.** Reconstruction PSNR corresponding to denoising using BDVP versus different DVP structures. (8 frames videos with additive Gaussian noise level of $\sigma = 25$)

