# OpenReview forum: "Untrained Neural Nets for Snapshot Compressive Imaging: Theory and Algorithms"
_NeurIPS.cc/2024/Conference — NeurIPS 2024 poster_

### Official Review · Reviewer_ytyT · 2024-07-08

**Soundness:** 3
**Presentation:** 3
**Contribution:** 3
**Rating:** 6
**Confidence:** 3

**Summary:**

Please see Strengths and Weaknesses.

**Strengths:**

1. Rigorous theoretical analysis of the proposed formulation
2. Detailed empirical evaluation.

**Weaknesses:**

1. Formatting in Figure 2 can be improved. The text near smaller cubes is not readable.

**Questions:**

NA

---

> ### Author Rebuttal · Authors · 2024-08-07
>
> We thank the reviewer for carefully reading our paper and providing constructive feedback.
>
> &nbsp;
>
> `Q-1:` Formatting in Figure 2 can be improved. The text near smaller cubes is not readable.
>
> `A-1:` We thanks the reviewer for the  suggestion.  In the revised version, we modify Figure 2 as follows: i) We increase the font size to make the figure more readable; ii) We add a detailed caption to explain the various components of the figure, including the meaning of different colored lines and dots; iii) We separate the 2D measurement $\bf y$ and the 3D binary mask $H$ to avoid confusion; iv) We change some of the solid lines to dashed lines to make the figure easier to interpret; v) We improve the layout of the figure to make it easier to follow. The updated Figure 2 is included in the one-page PDF response under the general rebuttal.

---

> > ### Comment · Reviewer_ytyT · 2024-08-14
> > **Acknowledging the rebuttal**
> >
> > Thank you for the rebuttal. I have no further questions.

---

### Official Review · Reviewer_E4xU · 2024-07-08

**Soundness:** 2
**Presentation:** 2
**Contribution:** 3
**Rating:** 5
**Confidence:** 4

**Summary:**

This work gives some theoretical analysis for mask optimization and DIP-based SCI recovery methods. The work claims that the proposed SCI-BDVP achieves SOTA performance among UNN methods.

**Strengths:**

1. This work provides some theoretical analysis, compared with the conventional work is rare.
2. Bagged DIP is introduced to develop SCI iterative methods and the proposed method claims to achieve SOTA performance among UNN methods.

**Weaknesses:**

1. While this work does some theoretical analysis by introducing DIP's theory, it cannot be directly generalized to all untrained networks. On the one hand, it seems impossible that all untrained networks satisfy the DIP hypothesis in the paper, and on the other hand, how can the UNNs used in this work be guaranteed to satisfy the DIP hypothesis and Lipschitz's condition? As far as I know, networks that satisfy Lipschitz's condition require a specific design, and this work does not seem to give some related introduction.
2. The core of this work is to prove theoretical results for untrained networks, but the actual contribution is less than stated. When the properties of certain untrained networks are demonstrated by DIP, the work does not propose any new hypotheses in snapshot compressive imaging but simply adopts the original one, which decreases the contribution of the paper.
3. When the DIP hypothesis is used to simplify the UNNs as a minimization operator, the work is merely proving a boundary based on the minimization operator. Similar theoretical results based on the minimization operator can also be found in the theoretical analysis section of GAP-Net[1].
4. The DIP hypothesis demonstrated in this paper seems to ignore some conditions in the original DIP paper and doesn't capture the unique characteristics of untrained networks, what if the same is true of pre-trained networks?
5. The comparison algorithm needs to be perfected. The current comparison algorithms seem to be some traditional optimization algorithms and your methods, and we even can't distinguish which of them is previous UNN works. So what does it mean that your method achieves SOTA performance in UNNs？
6. Considering this work as a deep network-based approach, such comparisons are unfair. Why not add some recent self-supervised methods to demonstrate the effectiveness of the algorithm? In addition, despite claiming to be compared with the supervised algorithms in noisy scenarios, some recent algorithms, such as SCI3D[2] and EfficientSCI[3] have not been included.

[1]Deep Unfolding for Snapshot Compressive Imaging, in IJCV2023.

[2]Dense Deep Unfolding Network with 3D-CNN Prior for Snapshot CompressiveImaging, in ICCV2021.

[3]EfficientSCI: Densely Connected Network with Space-time Factorization for Large-scale Video Snapshot Compressive Imaging, in CVPR2023.

**Questions:**

How can the UNNs used in this work be guaranteed to satisfy the DIP hypothesis and Lipschitz's condition?

**Limitations:**

The authors provide some limitations, which is their future research.

---

> ### Author Rebuttal · Authors · 2024-08-07
>
> We thank the reviewer for carefully reading our paper and providing constructive feedback.
>
> `Q-1:` Application of DIP hypothesis to all untrained networks and the relevance of the Lipschitz condition.
>
> `A-1:` As mentioned by the reviewer, not all untrained neural nets (UNNs) satisfy the DIP hypothesis and this  is not the assumption in our paper. The assumption is that for any class of signals, there exist UNN structures that satisfy the DIP hypothesis. More specifically,  consider a class of signals denoted by $\mathcal{Q}\subset\mathbb{R}^n$. A UNN, modeled as $g_{\theta}(\bf u)$, satisfies the DIP-hypothesis if for any randomly generated $\bf u\in\mathbb{R}^N$ (generated according to some preset distribution), and any $\bf x\in\mathcal{Q}$, $\min_{\theta} ||g_{\theta}(\bf u) - \bf x||_2 \leq \delta,
> $, almost surely. Existence of such UNN structures for different class of signals $\mathcal{Q}$ is shown empirically. For example, deep image prior (DIP) and deep decoder (DD) are well-known UNN structures  satisfying this property for natural images.
>
> Regarding Lipschitz continuity, note that our UNN architecture comprises multiple hidden layers, each employing a linear convolution operator followed by a ReLU nonlinearity. The final layer consists of a linear operation and a sigmoid activation function. Given the compositional nature of these layers, it is evident that our UNN satisfies the Lipschitz condition. Notably, our theorem is applicable to any Lipschitz constant $L$.
>
> &nbsp;
>
> `Q-2:` Questions about the contributions of the paper.
>
> `A-2:` The focus of this work is on the _application_ of UNNs in solving SCI problem and developing i) a theoretical  understanding of the problem, and ii)  a robust UNN-based SCI solution. Specifically, here are our key contributions:
>
> 1. Theoretically characterizing the performance of SCI solutions that employ a generic UNN for recovery. The key implication of our theoretical characterizations is that it enables us to optimize the parameters of SCI systems' hardware, under both noise-free and noisy setups.
>
> 2. Proposing a novel unsupervised SCI solution for video SCI. We establish the effectiveness of our proposed solution through  extensive simulations.
>
>
> &nbsp;
>
> `Q-3:` Comparison between this work and the GAP-Net paper [1] and the potential overlap between the two papers.
>
> `A-3:` In this paper, we theoretically analyze the performance of DIP-SCI optimization defined as
>
> \begin{align}
> \hat{\bf x}=\arg\min ||\bf y-\bf H\bf c||_2,
> \end{align}
>
> subject to $\bf c=g_{\theta}(\bf u),\theta\in[0,1]^k$
>
> [1] _does not_ include any theoretical analysis of this optimization or any of its variants. Instead, [1] focuses on the convergence behavior of projected gradient descent and its variant, GAP, in solving a related compression-based optimization. Also note that in [1], the  entries of the masks are assumed to be i.i.d.~ Gaussian, which is inconsistent with masks used in practice. Instead, in our work, we consider an i.i.d.~Bern$(p)$ distribution for masks.
>
> &nbsp;
>
> `Q-4:` Conditions of DIP hypothesis compared to the original DIP paper and connections to pre-trained networks.
>
> Pre-trained networks and UNNs differ significantly in their operational basis. While pre-trained networks leverage extensive datasets to acquire knowledge, UNNs operate without requiring any training data. This fundamental distinction grants UNNs a clear advantage in scenarios with limited data, making them more adaptable to various applications. Consequently, although pre-trained networks can be integrated into our framework, the unique strength of UNNs in data-scarce environments underscores their important role in our methodology.
>
> Finally, we would like to ask the reviewer to explicitly identify which DIP conditions they believe are overlooked in our paper. This helps us better address the concerns.
>
> &nbsp;
>
> `Q-5:` Comparison algorithms included in the paper.
>
> `A-5:` In Section 5, we compare the performance of our proposed method against several other  methods, listed in the section labeled `Datasets and baselines'. To the best of our knowledge, we cover all existing UNN-based video SCI solutions in our comparisons. We also extended one UNN-based method, originally proposed for hyperspectral SCI, and compared its performance against our method.
>
> To better clarify and distinguish the methods compared to our proposed method, we will update Table 1 in our main paper by organizing the comparisons into four categories:
>
> 1. GAP-TV [9], a traditional optimization-based algorithm.
>
> 2. PnP-FFDnet [17] and PnP-FastDVDnet [18], pre-trained deep denoisers + PnP algorithm.
>
> 3. Existing UNN-based solutions for video SCI [30], [31].
>
> 4. Our main proposed method (SCI-BDVP) and two simpler variants for comparison: 1) SCI-DVP, simple E2E DVP, 2) SCI-BDVP,  bagged E2E DVP, and 3) SCI-BDVP, our main proposed method.
>
> The last two categories are UNN-based methods, we consider our model to be state-of-the-art among this category.
>
> &nbsp;
>
> `Q-6:` Comparison with some mentioned recent papers.
>
> `A-6:` Methods such as SCI3D and EfficientSCI are supervised approaches that require extensive training data. A key limitation of these supervised methods is that they fix the sensing matrix during training. This makes it challenging to infer the impact of masks on performance and to separate this effect from the optimization performance. Additionally, applying the trained network to different sensing matrices and simulation settings typically results in performance degradation.
>
> In this paper, one of our main goals is to develop _unsupervised_ solutions that do not require training data or rely on prior knowledge of the sensing matrix and also to provide a fundamental understanding of the role of masks in performance. Therefore, we have not included supervised, mask-dependent methods in our comparisons.
>
> __References can be found in our main paper.__

---

> > ### Comment · Reviewer_E4xU · 2024-08-13
> >
> > Thank you for addressing my concerns. This work does have some unique theoretical contributions for snapshot imaging, thus, I tend to raise my score to borderline accept when some contents need to be further considered.
> > 1. The title and some sentences seem somewhat misleading, and it can easily be misinterpreted as suggesting that the proposed theory is for all UNNs rather than partial UNNs.
> > 2. If the proposed network has no constraints on linear weights, the Lipschitz condition is not necessarily satisfied when it's probably unbounded. In addition, considering activation functions and some special operators, other UNNs that satisfy the DIP hypothesis may also not satisfy the Lipschitz condition.
> > 3.  DIP conditions. The untrained neural network of the original DIP paper is initialized randomly. The pre-trained networks usually have some specific weight distribution and are subject to certain constraints. The current DIP assumption does not seem to account for the distribution of the parameter $\theta$.
> >
> > Note that one class of comparison methods in Table 3 is 'learning-based supervised methods', which should not be in the comparison when the author stated there is no comparison of supervised approaches.

---

> > > ### Author Response · Authors · 2024-08-14
> > >
> > > We appreciate the reviewer's response and comments. To address the concerns raised, note that:
> > >
> > > 1. As noted in our earlier response, similar to other UNN constructions in the literature, we utilize ReLU and Sigmoid activation functions, which satisfy Lipschitz continuity. We acknowledge that certain activation functions, such as the sign function, do not satisfy Lipschitz continuity. However, these activation functions are uncommon in practice due to the issues they pose for backpropagation and training. Additionally, unbounded weights, which can lead to an unbounded Lipschitz constant, are indeed undesirable as they may result in network instability. In the revised version, we will include a remark to discuss these points.
> > >
> > > 2. Regarding the DIP conditions and the initialization of weights in UNNs, please note that the initialization of weights in training UNNs primarily affects the convergence of the training algorithms and has no impact on our theoretical results. More importantly, different initialization methods, regardless of their distributions, all fall within the framework we have proposed. In our simulations, we initialize the weights using _Kaiming_ initialization, uniformly at random.
> > >
> > > 3. In Table 3, we have included some supervised methods because they utilize the exact same gradient step as our unsupervised methods but with different pre-trained projection modules. The results further demonstrate the effectiveness and generalization ability of our UNN-based model. Additionally, the key advantage of these methods lies in their flexible design, enabled by the iterative PnP approach, making them well-suited for studying the impact of masks on performance.

---

### Official Review · Reviewer_tX5M · 2024-07-10

**Soundness:** 4
**Presentation:** 4
**Contribution:** 4
**Rating:** 7
**Confidence:** 3

**Summary:**

The focus of this paper is on developing recovery algorithms of snapshot compressive imaging (SCI) using untrained neural networks (UNNs). Besides, the paper introduces the concept of bagged-deep-image-prior (bagged-DIP) to create SCI Bagged Deep Video Prior (SCI-BDVP) algorithms, which are designed to address common challenges faced by standard UNN solutions in SCI recovery. Extensive experiments demonstrate the effectiveness of the proposed method in video SCI recovery. In scenarios with noisy measurements, this untrained network even outperforms supervised methods.

**Strengths:**

- This paper is well structured and has clear logic.

- The experimental results on the performance are convincing.

- The proposed method is based on untrained neural network, making it more flexible to be applied in various scenes.

**Weaknesses:**

- For untrained methods, runtime may be a crucial metric [1,2]. But this paper doesn't provide analysis in comparison to existing methods.

- To improve clarity, consider adding clear notations for the symbols used. For example, what do different colored arrows represent in Figure 2, and the color of "Fusion" and "MSE" are too similar to be distinguished.

[1] Rui, Xiangyu, et al. "Unsupervised hyperspectral pansharpening via low-rank diffusion model." Information Fusion 107 (2024): 102325.
[2] Pang, Li, et al. "HIR-Diff: Unsupervised Hyperspectral Image Restoration Via Improved Diffusion Models." Proceedings of the IEEE/CVF Conference on Computer Vision and Pattern Recognition. 2024.

**Questions:**

- SCI is also widely used in hyperspectral image, can your method be used in hyperspectral reconstruction task? If so, what's the prominent advantage of your method?

- How does the method handle different noise models beyond additive Gaussian noise?

**Limitations:**

- Exploring the application on additional datasets from different domains or varing conditions in the future may increase persuasiveness, e.g. hyperspectral data. And exploring the performance of the proposed method under different noise models would provide a more comprehensive evaluation of its robustness and general applicability.

---

> ### Author Rebuttal · Authors · 2024-08-07
>
> We thank the reviewer for carefully reading our paper and providing constructive feedback.
>
> &nbsp;
>
> `Q-1:` Discussion about computational complexity and runtime.
>
> `A-1:` We have made the following changes to the paper: In the main body of the paper, in Section 5.1, we have added the following explanation on the computational complexity of our proposed method. _"The proposed SCI-BDVP method relies on the bagging of multiple DIP projections. These DIP projections, which vary depending on the patch size, involve different levels of computational complexity. Table 2 shows the average time required to perform each DIP projection for each patch size. As observed, the time increases considerably as the patch size decreases. This is expected because the number of networks that need to be trained grows significantly. (Refer to Figure 1 for a pictorial representation.)  Additional  computational complexity analysis  of our proposed method and its comparisons with other methods is included in Appendix B.3."_
>
> Furthermore, we have added Section B.3 in the Appendix, which provides detailed information about the computational and time complexity of our proposed method. It also includes a comparison between our method and other UNN-based approaches. For completeness, we have copied the newly added section here as well.
>
> _"Implementing SCI-BDVP involves outer loop iterations (described in Algorithm 1 in appendix B.1) and also inner loop iterations for training DIPs. Table 3 presents average number of inner loop iterations used for different patch sizes (64, 128, 256) of various videos, and the number of outer loop iterations. Detailed time consumption for each patch level computation is recorded in Table 2. A comparison across different UNN-based methods is provided in Table 1. All comparisons are performed on a single NVIDIA RTX 4090. It is important to note that training a bagged DIP requires training multiple separate DIPs. This process can be readily parallelized, which is expected to significantly speed up the algorithm. We plan to explore this direction to optimize the algorithm's efficiency in future work. Lastly, making a direct comparison among all methods is challenging because, for supervised methods, the main time is spent in training, whereas, for unsupervised methods, the main time is spent on training the UNNs. This is an expected trade-off for requiring no training data and achieving a robust solution."_
>
>
> ||**Methods**|**Time (min.)**|
> |-|-|-|
> |**No noise**|PnP-DIP| 18|
> ||Factorized-DVP|15|
> ||Simple-DVP (E2E)|10|
> ||SCI-BDVP|35 or 220|
> | **With noise**|PnP-DIP| 18|
> ||Factorized-DVP|$-$|
> ||Simple-DVP (E2E)|10|
> ||SCI-BDVP|40|
>
> **Table 1**: Time complexity of different methods on one 8-frame benchmark video.
>
> |**Patch size**|# of patches|Time (min.)|
> |-|-|-|
> |64|16|1.2|
> |128|4|0.28|
> |256|1|0.15|
>
> **Table 2:** Time complexity of our proposed SCI-BDVP was evaluated on various patch sizes (64, 128, 256) of video blocks, using a standard 1000 DVP iterations for training.
>
> &nbsp;
>
> `Q-2:` Clarity of notations and figure presentation.
>
> `A-2:` In the revised version, we modify Figure 2 as follows: i) increase the font size; ii)  add a detailed caption; iii)  separate the 2D measurement $\bf y$ and the 3D binary mask $H$; iv)  change some of the solid lines to dashed lines; v)  improve the layout of the figure to make it easier to follow. The updated Figure 2 is included in the one-page PDF response under the general rebuttal.
>
> Also, we have reviewed the paper to ensure that all notations are clearly defined and consistently used.
>
> &nbsp;
>
> `Q-3:` Potential application of  our proposed method in hyperspectral imaging.
>
> `A-3:` We expect our method to be applicable to hyperspectral imaging as well. In fact, one of the papers we have cited and used in our simulations for comparison (reference [27] in the paper) utilizes DIP for hyperspectral imaging (HI). Here are two key advantages of our proposed framework:
>
> 1. Our proposed theoretical framework is applicable to hyperspectral imaging and provides a solid theoretical foundation to analytically explore other aspects of HI, such as the effect of shifted masks.
>
> 2. A well-known challenge in using UNNs in reconstruction algorithms is their tendency to suffer from overfitting. However, our proposed method, based on bagging, is expected to perform well in HI and overcome these issues.
>
> To highlight this direction and its potential, we add this paragraph in the conclusion section of the revised version: "An important application of SCI is hyperspectral snapshot imaging (HSI). Our results in this paper provide a theoretical foundation to understand HSI systems and optimize their hardware. Additionally, the developed theoretical framework can be used to explore aspects specific to HSI, such as masks being shifted versions of each other. We also expect our algorithm to effectively address overfitting in HSI tasks, enhancing reconstruction performance. We plan to explore these aspects further in our future research."
>
> &nbsp;
>
> `Q-4:` Different noise models beyond additive Gaussian noise?
>
> `A-4:` The exploration of noise models beyond additive white Gaussian  (AWGN) noise is a valuable and important   direction for future work. While AWGN is a commonly-adopted model in many imaging solutions, including most SCI systems, there are situations, such as some newly-proposed coherence imaging SCI methods, that other types of noise such as speckle noise are dominant. We believe that  our UNN-based approach to model the source has the potential to address these alternative noise scenarios. To accommodate non-AWGN noise models, the loss function $||\bf y-\bf H\bf c||_2$ would require modification depending on the noise model. We consider this an important avenue for future investigation.

---

> > ### Comment · Reviewer_tX5M · 2024-08-09
> >
> > Thanks for your rebuttal. I have read the rebuttal and have no further questions.

---

### Official Review · Reviewer_pMi8 · 2024-07-23

**Soundness:** 4
**Presentation:** 3
**Contribution:** 3
**Rating:** 6
**Confidence:** 4

**Summary:**

This paper leverages untrained neural networks UNN (deep image priors DIP or deep decoder DD) as a prior to solve Snapshot Compressive Imaging (SCI), a technique used in ($n_1$ x $n_2$ x B)-dimensional 3-D imaging where the captured measurements lie in a 2-D plane ($n_1$ x $n_2$). The application of UNNs in the context of SCI itself is not novel; this paper's main contributions are:
1) Theoretical recovery guarantees for SCI (i.e. existence of a minimizer to the reconstruction problem) denoting number of 2-D frames B that can be recovered using 1 2-D measurement as a function of UNN model complexity (assuming original signal is close to range of UNN by $\delta$) under noise-free and additive gaussian noise settings.
2) Use bagged-DIP as algorithm for signal recovery, called SCI Bagged Deep Video Prior (BDVP).
3) Optimize binary-valued masks used in the measurement process and show empirical analysis.

**Strengths:**

The main contribution of this paper are theoretical reconstruction guarantees for SCI using UNNs under both noisy and noiseless measurements. They derive a bound on reconstruction error in terms of signal parameter B, Bernoulli sampling pattern mask p, measurement and signal parameter n, and $\sigma_z$ noise.

Authors further validate their reconstruction error bound and its dependence on sampling parameter p, on empirical datasets on 6 videos: Drop, Runner, Aerial, Crash, Kobe, Traffic.

They propose SCI Bagged Deep Video Prior method as an algorithmic framework for solving SCI. On the empirical data, they show improved SNR and SSIM against baseline untrained methods.

**Weaknesses:**

Main comments:
Computational complexity: Deep Video Prior/Untrained Network Prior setups have high computational complexity; on top of this a bagging approach requires solving K such problems simultaneously. This requires a discussion on how computational complexity compares to baselines.

Theoretical claims are summarized in main paper; however I was not able to find longer version of paper in supplementary zip folder to validate the proofs.

Theorem 3.1: Are parameters $p$ in $u [0,1]^p$ and Bern($p$) for $D_{i,j}$ the same?

Minor comments:
line 47-48: "Existing UNN-based SCI solutions either recover the image 48 end-to-end in one shot or employ iterative methods akin to projected gradient descent (PGD)." (relevant citations missing)

line 88-97: Literature review of DIP + UNN: Add relevant citation to "Qiao, M., Liu, X., & Yuan, X. (2021). Snapshot temporal compressive microscopy using an iterative algorithm with untrained neural networks. Optics Letters, 46(8), 1888-1891."

Theorem 3.1. reconstruction error bound - $\rho$ not defined.

**Questions:**

Theoretical claims are summarized in main paper; however I was not able to find longer version of paper in supplementary zip folder to validate the proofs - this would help consolidate the contents.

**Limitations:**

Limitations have been discussed; they should additionally discuss time/computational complexity of the proposed BDVP algorithm.

---

> ### Author Rebuttal · Authors · 2024-08-07
>
> We thank the reviewer for carefully reading our paper and providing constructive feedback.
>
> &nbsp;
>
> `Q-1:` Discussion about Computational complexity.
>
> `A-1:` We have made the following changes to the paper: In the main body of the paper, in Section 5.1, we have added the following explanation on the computational complexity of our proposed method. _"The proposed SCI-BDVP method relies on the bagging of multiple DIP projections. These DIP projections, which vary depending on the patch size, involve different levels of computational complexity. Table 2 shows the average time required to perform each DIP projection for each patch size. As observed, the time increases considerably as the patch size decreases. This is expected because the number of networks that need to be trained grows significantly. (Refer to Figure 1 for a pictorial representation.)  Additional  computational complexity analysis  of our proposed method and its comparisons with other methods is included in Appendix B.3."_
>
> Furthermore, we have added Section B.3 in the Appendix, which provides detailed information about the computational and time complexity of our proposed method. It also includes a comparison between our method and other UNN-based approaches. For completeness, we have copied the newly added section here as well.
>
> _"Implementing SCI-BDVP involves outer loop iterations (described in Algorithm 1 in appendix B.1) and also inner loop iterations for training DIPs. Table 3 presents average number of inner loop iterations used for different patch sizes (64, 128, 256) of various videos, and the number of outer loop iterations. Detailed time consumption for each patch level computation is recorded in Table 2. A comparison across different UNN-based methods is provided in Table 1. All comparisons are performed on a single NVIDIA RTX 4090. It is important to note that training a bagged DIP requires training multiple separate DIPs. This process can be readily parallelized, which is expected to significantly speed up the algorithm. We plan to explore this direction to optimize the algorithm's efficiency in future work. Lastly, making a direct comparison among all methods is challenging because, for supervised methods, the main time is spent in training, whereas, for unsupervised methods, the main time is spent on training the UNNs. This is an expected trade-off for requiring no training data and achieving a robust solution."_
>
> || **Methods**| **Time (min.)** |
> |-|-|-|
> | **No noise**| PnP-DIP| 18|
> || Factorized-DVP| 15|
> || Simple-DVP (E2E)| 10|
> || SCI-BDVP| 35 or 220|
> | **With noise**| PnP-DIP| 18|
> || Factorized-DVP| $-$|
> || Simple-DVP (E2E)| 10|
> || SCI-BDVP|40|
>
> **Table 1**: Time complexity of different methods on one 8-frame benchmark video.
>
> | **Patch size** | # of patches | Time (min.) |
> |-|-|-|
> | 64| 16| 1.2|
> | 128| 4| 0.28|
> | 256| 1| 0.15|
>
> **Table 2:** Time complexity of our proposed SCI-BDVP was evaluated on various patch sizes (64, 128, 256) of video blocks, using a standard 1000 DVP iterations for training.
>
> &nbsp;
>
> `Q-2:` Proofs of the theoretical results.
>
> `A-2:` The main pdf file (29 pages) includes detailed proofs of all the results. More specifically, in Appendix A, we have provided detailed proofs in the following order: proof of Theorem 3.1 in A.2, proof for Corollary 3.3 in A.3, proof for Theorem 3.4 in A.4 and finally proof for Theorem 3.5 in A.5.
>
> &nbsp;
>
> `Q-3:` Theorem 3.1: Are parameters $p$ in $u[0,1]^p$ and Bern($p$) for $D_{i,j}$ the same?
>
> `A-3:` We had inadvertently overloaded the symbol $p$. We will substitute $p$ in $u[0,1]^p$ with $N$.
>
> &nbsp;
>
> `Q-4:` Theorem 3.1. reconstruction error bound - $\rho$ not defined.
>
> `A-4:` Thanks for pointing this out. $\rho$ denotes an upper bound on the $\ell$-infinity norm of the signals in $\mathcal{Q}$. We  added the description in the revised version.

---

> > ### Comment · Reviewer_pMi8 · 2024-08-14
> > **Thanks for addressing concerns. No further comments.**
> >
> > Concerns have been addressed adequately.

---

### Author Rebuttal · Authors · 2024-08-07

We thank all reviewers for their valuable feedback and thoughtful comments. We have carefully considered their suggestions and revised the paper accordingly. In the following, we address the main comments/questions from each reviewer..

The attached pdf file includes an improved version of Figure 2 in our original manuscript.

---

### Decision · Program_Chairs · 2024-09-25

**Decision:**

Accept (poster)

**Comment:**

The paper introduces a new approach for solving the snapshot compressive imaging (SCI) problem, where the goal is to reconstruct 3D data cubes from 2D measurements. Applications of this problem include video and hyperspectral imaging. The approach is based on a 3D extension of bagged deep image priors. The authors provide both theoretical bounds on the reconstruction error (under the so-called deep image prior hypothesis), as well as simulation results on benchmark video datasets.

The paper is well written, the method is sensible, and the supporting results are comprehensive. During the response phase the authors were able to adequately address most of the reviewers' comments.